# Development of Methods for Remote Monitoring of Leaf Diseases in Wheat Agrocenoses

**DOI:** 10.3390/plants12183223

**Published:** 2023-09-10

**Authors:** Igor Sereda, Roman Danilov, Oksana Kremneva, Mikhail Zimin, Yuri Podushin

**Affiliations:** 1Faculty of Geography, Lomonosov Moscow State University, 119991 Moscow, Russia; iisereda@mail.ru (I.S.); ziminmv@mail.ru (M.Z.); 2Federal State Budgetary Scientific Institution “Federal Research Center of Biological Plant Protection” (FSBSI FRCBPP), 350039 Krasnodar, Russia; 3Federal State Budgetary Educational Institution “Kuban State Agrarian University Named after I.T. Trubilin”, 350004 Krasnodar, Russia; yrapod@mail.ru

**Keywords:** wheat, pathogens, phytosanitary monitoring, earth remote sensing, hyperspectral analysis, GIS, spectrometry, spore catcher

## Abstract

The development of remote methods for diagnosing the state of crops using spectral equipment for remote sensing of the Earth and original monitoring tools is the most promising solution to the problem of monitoring diseases of wheat agrocenoses. A research site was created on the experimental field of the Federal Research Center of Biological Plant Protection. Within the experimental field with a total area of 1 ha, test plots were allocated to create an artificial infectious background, and the corresponding control plots were treated with fungicides. The research methodology is based on the time synchronization of high-precision ground-based spectrometric measurements with satellite and unmanned remote surveys and the comparison of the obtained data with phytopathological field surveys. Our results show that the least-affected plants predominantly had lower reflectance values in the green, red, and red-edge spectral ranges and high values in the near-infrared range throughout the growing season. The most informative spectral ranges when using satellite images and multispectral cameras placed on UAVs are the red and IR ranges. At the same time, the high frequency of measurements is of key importance for determining the level of pathogenic background. We conclude that information acquisition density does not play as significant of a role as the repetition of measurements when carrying out ground-based spectrometry. The use of vegetation indices in assessing the dynamics of the spectral images of various survey systems allows us to bring them to similar values.

## 1. Introduction

Wheat is the main worldwide staple food. Economically important fungal diseases of wheat are Fusarium head blight (FHB) (*Fusarium graminearum* Schwabe.), brown and yellow rust (*Puccinia triticina* Erikss., *Puccinia striiformis* West.), yellow spot (*Pyrenophora tritici-repentis* (Died.) Drechsler), septoria (*Septoria* spp.), and powdery mildew (*Blumeria graminis* (DC.) Speer). Diseases are very harmful and widespread around the world [1,2] and in Russia, especially in its southern region [3,4]. The FAO and UN estimate that about 10% of yield is lost to pathogens in developed countries and 20–50% in developing ones [5].

Effective phytosanitary monitoring is a critical component of pest management programs. It allows for the timely detection of disease development in crops [6]. Today in intensive agriculture, traditional phytosanitary monitoring is hampered by the presence of large sown areas. Hence, there is a lack of proper control by specialists. The most promising way to solve this problem is the development of remote methods of crop diagnostics, based on the use of remote sensing equipment and original monitoring tools [7,8].

Nowadays, hyperspectral equipment is widely used. It allows obtaining data of high spatial and spectral resolution [9,10,11,12,13]. There are positive results of studies on the development of remote methods for detecting FHB (*F. graminearum* Schwabe.) [14,15,16,17], yellow rust (*P. striiformis* West.) [18,19,20,21], powdery mildew (*B. graminis* (DC.) Speer) [22], and other wheat diseases based on the analysis of hyperspectral data.

However, despite the impressive research results, the actual application of hyperspectral technologies for monitoring crop diseases is still significantly hindered by a number of unsolved practical, scientific, and methodological problems [10,12,13]. First is the high cost of hyperspectral data, as well as the complexity of their processing. Another problem is the difficulty of taking into account the mutual influence of environmental factors (soil type, weather conditions, illumination of the Earth’s surface, etc.), which determine the conditions for obtaining experimental data within a particular research region. The influence of these factors on the spectral characteristics of crops can be non-uniform during the growing season. This requires a significant transformation of models created in specific conditions when applied to a new region [13,23,24,25].

A great obstacle is the issue of early diagnostics of the development of pathogens, as well as the accurate detection of a specific disease in the field, when there is a simultaneous influence of several stress factors that can cause similar changes in the spectral characteristics of the objects under study [9,26,27,28,29]. The difference in the spectral responses of different varieties of the same crop also presents significant difficulties. Therefore, the model for diagnosing the development of the disease in one particular wheat variety may not be applicable to another variety [14,30,31]. Finally, there is a need to study the dynamics of changes in the spectral images of cultivated crops against the background of the development of diseases over time during their growing season [10].

Some researchers assume that a big gap in the scientific basis for planning experiments on the use of Earth remote sensing data to determine the state of plants is associated with the lack of sufficient interaction between specialists in the field of technology and biology [10,13]. Therefore, it is necessary to study the possibilities of combining high-resolution satellite data with the results of surveys obtained using UAVs and ground-based observations, taking into account the biological aspects of the experiment to build a consistent time series of remote sensing data [10].

For instance, in Serrano [32]’s research, moisture content, crude protein, and other characteristics of dryland pastures were evaluated with the NDVI index obtained using a proximal optical sensor and satellite images. The correlation coefficients showed moderately high accuracy between the spectral data of vegetation and vegetation conditions. Bausch and Khosla [33] determined the most informative vegetation index for the nitrogen content estimation in maize according to ground-based sensor data and Quick-Bird images. It was found that normalized GNDVI is the most correct index for monitoring nitrogen using satellite images. Mezera et al. [34] evaluated the relationship between nitrogen and vegetation indices on the basis of proximal and remote sensing methods of Sentinel-2 long-term satellite images. The study showed that vegetation indices show a high correlation with nitrogen content, but their values vary greatly depending on the season. Stettmer et al. [35] evaluated the dependence of nitrogen content on the spectral characteristics of plants at various phenophases of winter wheat. It was shown that for all key stages (GS 31, 39, 55, 65), the correlation remains high. Also, to improve the accuracy of the assessment, it was recommended to use the higher spatial resolution of satellite data (5 m^2^).

Nevertheless, most studies were aimed at finding connections between the spectral characteristics of the culture and its macroelements, primarily nitrogen. Much less attention was paid to the study of crop diseases and, in particular, their early detection.

One supposes that crop-disease-monitoring systems should take into account the specific features of their occurrence and development. It is known that the number of spores of phytopathogenic fungi found in the air above the affected crops is an important indicator of the epiphytotic potential of the pathogen [32,33]. Studies by Russian and foreign researchers devoted to the phytosanitary monitoring of crops using spore-catching equipment indicate the possibility of early (5–7 days before the onset of visible symptoms) detection of spores of the causative agent of rust fungal diseases in wheat crops [36,37,38,39].

Studies on the aerobiology of wheat fungal diseases [40,41] have shown a relationship between the infestation of crops and the number of uredospores of pathogens in the surface air layer. Uredospores of pathogens can be detected by special spore-catching devices installed on aircraft [42,43]. Thus, the ability to assess the degree of the disease development or even predict its development by the number of uredospores in the air will ensure the objectivity of the qualitative and quantitative interpretation of remote sensing data.

The data obtained from aerobiological sounding and spectral response measurements can complement each other, creating a kind of two-stage monitoring system.

Here, we aim to develop a methodological basis for the use of ground- and aerospace-based spectral equipment in combination with technical means for controlling phytopathogenic infection for the remote monitoring of wheat agrocenoses.

## 2. Results

Various approaches to obtaining spectral data of vegetation were applied: ground-based spectrometry, surveys using UAVs, and satellite imagery. Of all the above-mentioned methods, only satellite imagery has a complete and relatively continuous nature of data acquisition. In addition, these data fully cover the period of field examinations of the pathogenic background. Consequently, satellite imagery was taken as the main source of information for analysis. Unfortunately, the spatial resolution of these data does not allow for estimating the spectral characteristics of the small-plot experiment. To compare the data from different imaging systems with each other, the spectral brightness values were reduced to the system containing the smallest number of spectral ranges: the Parrot SEQUOIA + camera (Parrot, Paris, France) mounted on the UAV.

The study of the spectral characteristics of various survey systems showed that the highest SBC indicators are typical for aerial survey data. The lowest SBC indicators are standard for satellite imagery. At the same time, the use of vegetation indices, such as NDVI, makes it possible to bring different survey systems to similar values for most measurements.

Let us mention that the dynamics of the spectral characteristics of the satellite imagery correlate well with the aerial survey data, except for the range 712–722. This is due to differences in the spectral resolution of the corresponding data channel of the systems. Thus, we can make a preliminary conclusion that satellite images and UAV data can be interchangeable and complement each other in the remote monitoring of crops (Figure 1).

A careful consideration of the SBC dynamics at different wavelengths using satellite imagery as an example suggests that the control area with the absence of a pathogenic background has its own distinctive features. The spectral brightness of the control section is one of the lowest in the ranges of 550–570 nm, 663–673 nm, and 712–722 nm throughout the entire observation period. It is the highest at the final development stages, in which field inspections were not carried out. In the IR range (820–860 nm), the spectral brightness of the control area is one of the highest during the entire period of satellite observations. Its NDVI index is also one of the highest throughout the study. There is a tendency for a similar behavior of the spectral image curve for the plots that are less prone to diseases. However, a number of plots are inconsistent with this template (Figure 2). Presumably, this is due to the low contrast of the level of pathogenic background between different test plots (Figure 1).

Let us compare the values of the spectral image obtained using satellite imagery with the UAV imagery data only from 19 May to 3 June, when they were taken simultaneously (Figure 3). However, a similar trend is observed here as well: low values in the green, red, and red-edge ranges of the spectrum and high values of SBC in the IR range on the plots with a less developed pathogenic background.

The spectral data analysis in a small-plot experiment with crops of Yuka and Alekseich varieties revealed that the infectious backgrounds of crops of these varieties were characterized by reduced SBC values compared to the control plots in all considered ranges of the spectrum (Figure 4). Moreover, the height of plants and the content of chlorophyll had a positive correlation. Comparison of the spectral characteristics of the infected and control crops of the two studied varieties showed that the quantitative indicators of the SBC values of each variety were largely determined by their biometric features and, possibly, by the different nature of the occurring physiological processes caused by pathogen development.

Based on the results of data processing, correlation dependences of indicators of the degree of development of diseases with variable SBC values were obtained in different channels of the Planet Dove satellite system and the multispectral camera Parrot SEQUOIA + (Table 1 and Table 2).

According to Planet Dove data, in the first considered period of vegetation of winter wheat plants (GS 40–47 “flag leaf”), a positive and statistically significant correlation between the degree of powdery mildew development and SBC values in the spectral channel 490 nm, as well as negative correlations in the channel 565 nm and on the values of vegetation index NDVI2 were observed. This is explained by the onset of mass manifestation of disease presence, which was observed at each test plot. Such a statistically significant and rather high correlation of pathogen development with satellite imagery data indicates the potential possibility of its detection in winter wheat crops at the earliest stages of development. Septoriosis and yellow rust were also diagnosed on some plots.

In the second time period associated with the earing phase (GS 50–59 “heading”), the beginning of the complex manifestation of septoriosis and yellow rust was observed. According to Planet Dove data, a high and statistically significant positive correlation was observed for septoriosis and yellow rust in spectral channels 443 and 490 nm and a negative correlation for yellow spot in spectral channels 443 and 610 nm. According to UAV imagery data for powdery mildew, a high correlation was observed in spectral channels 550 and 660 nm, as well as for NDVI2 vegetation index values. Thus, satellite imagery and UAV data turned out to be complementary sources of information.

For the vegetation period of winter wheat development associated with the flowering phase (GS 60–70 “flowering”), there was a high positive statistically significant correlation of yellow rust development and NDVI2 values in the 443 nm Planet Dove spectral channel. UAV imagery data showed a high and statistically significant level of correlation of powdery mildew development indicators with NDVI2 vegetation index variables. The average level of correlation of the 550 nm spectral channel with yellow rust development was also shown.

During the milk-wax ripening phase (GS 71–82), a significant decrease in the correlation of powdery mildew development with satellite and drone imagery data was observed. This is explained by the cessation of pathogen development due to its biological cycle of transition to the resting phase. The maximum level of development was observed for septoriosis, which manifested itself in a positive statistically significant correlation with the variable values of vegetation index NDVI2, obtained from the data of the Planet Dove system. In addition, there was a statistically significant correlation of brown rust development with variable values of the 443 nm spectral channel, which is obviously associated with a new phase of its manifestation.

Using the PSL-3 device, we managed to fix the sporulation of almost all listed pathogens except septoriosis spot disease, since the spreading of this pathogen occurs exclusively with dripping moisture (rain, dew), and work with the device is carried out in dry weather.

In phase GS 40–47, “flag leaf”, due to the minimal manifestation of diseases and a single number of spores for most variants of the experiment, the relationship between the number of captured spores and the degree of disease development was insignificant or absent (Table 3). Positive and statistically significant dependence was revealed only for yellow rust disease with a high correlation coefficient equal to 0.9.

In phase GS 61, “early flowering”, the correlation analysis revealed quite high levels of statistically significant dependence of the number of trapped spores and the degree of powdery mildew and yellow rust development equal to 0.7–0.8. No correlation relationship was found for yellow spot disease.

Phase GS 71–82, “milk-wax ripeness”, also revealed a fairly high correlation level of disease development indicators and the number of powdery mildew spores, which was 0.7. No correlation relationship was found for yellow spot disease. A correlation coefficient with a high statistically significant value of 0.9 was obtained for yellow rust.

Thus, the influence of weather and the choice of varieties on the development and spread of fungal leaf diseases of wheat has been established and statistically proven. The obtained results indicate the potential possibility of using a device for determining the infestation of plants or other similar spore-catching equipment for compiling a predictive model for the development of pathogens.

The results of one-factor analysis of variance confirmed the influence of the disease development factor for all pathogens at a high level of statistical significance (Table 4).

Based on the results of posterior analysis, a statistically significant reliable difference between control plots and infected plots was found (Table 5, Table 6, Table 7 and Table 8). Plots with minimal development of powdery mildew (1.5%) were also singled out in a separate group. Groups of plots with a gradation of disease development (2.5% and 3.5–4%) were identical in spectral characteristics (Table 5).

For septoriosis and yellow rust, despite the significant gradation of their development, the differences were manifested only in comparison with control plots in spectral channels 490, 550, 620, 720, and 1445 nm (Table 6 and Table 7).

## 3. Discussion

Estimates of the physiological state of crops of cultivated plants based on the analysis of hyperspectral data are the subject of active study by many researchers [14,15,16,17,18,19,20,21,22]. Such analysis can be used to develop precision phytosanitary monitoring methods. It is noteworthy that a significant number of publications are devoted to the identification of wheat diseases [11,12,13]. Notable results have been obtained in the detection of diseases such as FHB (*F. graminearum* Schwabe) [14,15,16,17], yellow rust (*P. striiformis* West.) [18,19,20,21], powdery mildew (*Blumeria graminis* (DC.) Speer), and other wheat diseases. [22]. These studies implemented various data processing algorithms based on the methods of vegetation indexing, cluster and discriminant analysis, and machine and deep learning. The areas of application of hyper-spectral technologies in disease diagnosis also differ. The main objectives are early detection of pathogens, their identification and differentiation, assessment of the degree of disease, and assessment of the resistance of a variety of wheat genotypes to diseases [11].

The fundamental novelty of our study is a methodological approach based on the creation of test plots that allow synchronization of high-precision ground-based spectrometric measurements with satellite and unmanned remote surveys with a comparison of the obtained data with the results of phytopathological field surveys. This approach makes it possible to scale the developed technologies with the help of remote sensing monitoring systems. In addition, the proposed scheme for the integrated interpretation of ground and remote data is supplemented by aerobiological methods for controlling phytopathogenic infection through the use of special technical means.

According to the data of the Dove Planet satellite system, in the first growing season of winter wheat growth (GS 40–47 “flag leaf”), a statistically significant correlation was found between the indicators of the disease development and the variable SBC values of the 490 and 565 nm spectral channels, as well as the values of the vegetation index NDVI2 for the causative agent of powdery mildew (Table 1). Such a statistically significant and rather high correlation of pathogen development with satellite imagery data indicates the potential for its detection in winter wheat crops at the earliest stages of development. In addition, an average degree of correlation (0.70) was established between the degree of powdery mildew development and the number of pathogen spores detected using the PSL-3 air sampler (Table 3). Septoria was detected as a single occurrence in one of ten test plots. Thus, there was no significant correlation with satellite imagery data for this pathogen. It was hardly possible to detect the spores of this pathogen, since its spread occurs exclusively with droplet moisture (rain, dew), and the PSL-3 air sampler can only be operated in dry weather. There was no statistically significant correlation with the Dove Planet satellite system data for yellow rust as well. However, a statistically significant relationship between the degree of disease development and the number of pathogen spores detected using the PSL-3 air sampler was confirmed (Table 3). On the one hand, it helps to detect the pathogen in winter wheat crops at the early stages of its development, and on the other hand, it can serve as a basis for predicting its further development at subsequent stages of winter wheat ontogeny, taking into account the influence of weather factors.

In subsequent periods of winter wheat ontogeny (GS 50–59 “heading”, GS 60–70 “flowering”, GS 71–82 “milk-wax ripeness”), there were found statistically significant dependences of disease development and SBC values of the spectral channels of the Dove Planet satellite system data and the multispectral camera Parrot SEQUOIA + mounted on the UAV (Table 1 and Table 2). The nature of these dependencies for the two compared survey systems with different central values of the spectral channels was different. For example, in the GS 50–59 “heading” phase, according to the Dove Planet data, no statistically significant dependence of powdery mildew development on any of the spectral channels and values of vegetation indices was revealed. But the UAV data of the same time period showed an average (0.65–0.70) statistically significant level of correlation of this pathogen’s development with variable SBC values of the 550 and 565 nm spectral channels, as well as the values of the NDVI2 vegetation index (Table 2).

It can be concluded that satellite images and UAV data turned out to be complementary sources of information. On the other hand, for different pathogens, the direction of the revealed correlation dependencies also differed. So, for example, according to the Dove Planet data, in the GS 50–59 “heading” phase, the variable SBC values of the 443 nm spectral channel were characterized by a positive correlation with the development degree of yellow rust but a negative one with yellow spot (Table 1).

According to the multispectral camera Parrot SEQUOIA+ data in the same growing season, the variable SBC values of the 660 nm spectral channel showed a positive correlation for powdery mildew and a negative correlation for septoria (Table 2). The revealed patterns of correlation, their nature, and their direction can become a potential basis for identifying the ratio of the complex development of several pathogens in winter wheat crops. It should also be noted that for powdery mildew during the growing season, including the GS 50–59 “heading”, GS 60–70 “flowering”, and GS 71–82 “milk-wax ripeness” phases, there was established an average and statistically significant correlation dependence of disease development with the number of pathogen spores detected using the PSL-3 air sampler equal to 0.70 (Table 3). A high and statistically significant correlation level equal to 0.80–0.90 was found for yellow rust.

One-way analysis of variance confirmed the influence of the plant disease development factor on the spectral characteristics of the studied winter wheat crops (Table 4). It also identified statistically significant infected test sites from controls (Table 5, Table 6 and Table 7). These differences were most pronounced in the spectral channels of 490, 550, 660, and 720 nm, which are sensitive to the ratio of the content of chlorophylls A and B, carotenoids, and anthocyanins in plants, as well as the effects of stress factors. Data for one-way analysis of variance were obtained as a result of ground spectrometry of winter wheat crops in the GS 60–70 “flowering” phase. The degree of development of powdery mildew on infected test plots varied between 1.5 and 4.0%; of septoria, it varied between 1.0 and 2.5%; and of yellow rust, it varied between 2.5 and 10.0%. In the control plot, the average values were 0.62% for powdery mildew, 1.43% for septoria, and 0.09% for yellow rust. It is noteworthy that the most pronounced differentiation of spectral characteristics was revealed when grouping variable SBC values into categories corresponding to different degrees of powdery mildew development. Thus, infected areas with a minimum pathogen development of 1.5% were allocated to a separate group. The plot groups with a gradation of disease development of 2.5% and 3.5–4% were identical in terms of spectral characteristics (Table 5). This indirectly confirms the most pronounced effect of the degree of powdery mildew development on the spectral characteristics of the studied crops compared to other pathogens.

The low productivity of ground-based spectrometry relative to satellite and UAV data was noted. At the same time, the multitemporal dynamics of the spectral image of plants for each of these survey levels remain similar. In this regard, in such studies, it is recommended to focus on aerial and satellite-based data, especially if there is a task of scaling. However, since ground spectrometry is based on active sensors with their own light source and is therefore less dependent on weather, it should not be abandoned entirely. At the current stage, ground-based spectrometry is used as a method that allows one to study in detail the changes in the spectral characteristics of crops and over time in order to identify certain patterns. These data can be used to better interpret the results of space and unmanned surveys.

Based on the study, we can provide a number of recommendations for the further development of the experiment. First of all, it is recommended to abandon ground-based spectrometry, as its low efficiency was revealed when moving to large areas for monitoring. In addition, in the course of previous seasons, sufficient material was accumulated in this direction. At the same time, it is desirable to pay more attention to satellite imagery and surveys using unmanned aerial vehicles, since they allow surveying over a large area, with a qualitative level of data calibration; they are interchangeable, and also, presumably, less expensive analogs of spectrometric studies.

Satellite and aerial photography must be performed during the entire observation period from the moment of plant inoculation. Aerial photography must be carried out at least once every 5–7 days. Satellite remote sensing data are recommended to be booked in advance. It is also necessary to organize test plots in such a way as to achieve their maximum homogeneity, but noticeable differentiation in terms of the degree of development of the pathogenic background.

Understanding and systematizing the identified technological modes of using spectral equipment in combination with original technical means for pathogen development diagnostics, as well as the established statistically significant relationships between the dynamics of the spectral image of various survey systems and the development of the pathogenic background of winter wheat crops, will make it possible to formulate the methodological foundations of remote sensing monitoring of wheat agrocenoses.

## 4. Materials and Methods

### 4.1. Arrangement of Test Plots

The studies were carried out in the field conditions of the North Caucasus region of Russia, on the experimental fields of the Federal State Budgetary Scientific Institution “Federal Research Center for Biological Plant Protection” (FSBSI FRCBPP) in Krasnodar (45°2.413′0″ N. 38°58.5598′0” E, 29 m a.s.l. m.) during the growing season of 2022. According to the Koppen–Geiger classification, the climate of the study area is transitional from temperate continental to subtropical (Cfa) [44]. This region is characterized by long, hot summers and mild to moderately warm winters. Transitional seasons are weakly expressed. The average amount of precipitation per year is 700–750 mm. The average annual air temperature is +13.4 °C, and the average annual air humidity is 71%. In the growing season of 2022, the average daily temperatures were 12.2 °C in April, 17.5 °C in May, and 22.6 °C in June; the average rainfall was 23 mm in April, 54 mm in May, and 159 mm in June. Air humidity varied from 65 to 70%. The soil cover of the territory is represented by leached low-humus chernozems [45].

The research site was represented by sowing winter wheat of the promising Alekseich variety. Ten test plots were allocated to create an artificial infectious background. Accordingly, there were 10 control plots to ensure the comparability of remote sensing survey data with the results of ground-based spectrometric measurements within the experimental field with a total area of 1 ha. The size of each test plot was 10 m × 10 m (100 m^2^) (Figure 5A,B). In addition to the main test plots, a small-plot experiment was established with Alekseich and Yuka varieties. The purpose of this experiment was to study in detail the reflectivity of winter wheat varieties, which are characterized by different degrees of resistance to phytopathogens and different biometric parameters. The small plots were divided into infectious background and control plots (Figure 5C). Artificial infection of plants was carried out on infected plots. The control plots were treated with fungicides to suppress the development of pathogens and create a relatively clean background. The rest of the field was not subject to accounting; its examination was not carried out. The studies were carried out on dedicated and controlled test sites. The remaining field area represents the background areas, which avoid errors associated with field edge effects.

To create an infectious background on selected test plots, the method of artificial infection of winter wheat plants with spores of phytopathogens was used. Infection of winter wheat plants was carried out on 16 April in the “beginning of the tube” phase (GS 30-32). A 1:100 mixture of urediniospores with talc was used for plant inoculation at a loading of 5 mg spores/m^2^. The creation of a clean background (without diseases) was carried out by 2-fold treatment of the control plots with the systemic fungicide Falcon. EC: 1st treatment on 25 April 2022 (flag list phase), 2nd treatment on 9 May 2022 (phase “early flowering” GS 61).

The research methodology was based on time synchronization of high-precision ground-based spectrometric measurements with satellite and unmanned remote surveys and comparison of the obtained data with the results of phytopathological field surveys (Table 8).

Three key moments of vegetation timed to specific stages of ontogenesis of winter wheat plants were identified:The first key time period of vegetation is timed to the end of April/beginning of May, when wheat plants reach the GS 40–47 “flag leaf” phase. This time point is interesting for the possibility of early detection of infectious onset at the initial stages of pathogen development.The second growing period refers to the second week of May and includes the phases of “earing and beginning of flowering” (GS 50–70). This time is associated with intensive manifestation of all leaf-stem diseases. This time point is an important link for making a predictive model of pathogen development as it allows us to make a comparative analysis of quantitative indicators of their development (degree of development, number of spores) from the moment of primary signs (after the incubation period) to the beginning of intensive manifestation.The third period under consideration refers to the phase GS 71–82 of milk-wax ripeness of winter wheat and comes at the end of May/beginning of June. The data obtained now are the final logical component of the research and can be a potential basis for yield forecasting.

### 4.2. Field Inspections

The assessment of the degree of disease development was based on the visual counting of the ratio of the proportion of the affected area of the plant leaf lamina to its total area. Visual counts of winter wheat disease development were carried out along the diagonal of each plot with an area of 10 m^2^. During the surveys, 30 plants were selected, and then for each tier (first, second leaf, etc.), according to international scales, the percentage of leaf lesions was given. The Peterson scale [46] was used to assess the degree of rust disease damage, the modified Saari and Prescot scale [47] was used to assess the degree of pyrenophorosis damage, and special scales developed by CIMMYT [48,49] were used to assess the degree of powdery mildew, spot blight, and septoriosis damage.

For each test plot, the average indices of the degree of disease development were calculated according to Formula (1) (Table 9 and Table 10):(1)$$R=\frac{1}{n}\sum_{i=1}^{n}r_{i}$$
where *R* is the average degree of development of the disease, %; *r* is the degree of development of the disease of an individual plant, %; *n* is the total number of registered plants, pcs.

Parallel to the accounting of the degree of disease development, air sampling over winter wheat crops was carried out using the original air sampler PSL-2 developed at FSBSI FRCBPP [50]. The device is an impactor, inside which there is a slide with the initial size of the composition (vaseline) in which the spores of phytopathogenic fungi are deposited. Sampling was conducted along the diagonal of each plot at five points. The sampling time was one minute. To detect, identify, and quantify phytopathogenic fungi, the samples were examined under a light microscope at 10x objective magnification.

### 4.3. Ground Spectrometry

Ground-based spectrometry was carried out non-contact at a height of 1.2–1.4 m from the Earth’s surface in the electromagnetic radiation range from 350 to 2500 nm with a spectral resolution of 1–10 nm. To this end, we used the ASD FieldSpec 3 Hi-Res spectroradiometer (ASD, Boulder, CO, USA) [51] designed to measure the absolute and relative values of the radiance. We carried out measurements in clear sunny weather with a minimum amount of clouds at sun heights of more than 35° to ensure the comparability of the obtained data. Under such conditions, lighting conditions change much less, which reduces the error associated with the influence of this factor. Vegetation measurements in the small-plot experiment were carried out in two series of five repetitions which were interrupted by measurements of the calibration white panel. This measure was taken to reduce the influence of the uneven lighting factor. For each plot of the main experiment, 30 measurements of the vegetation cover were carried out as well as measurements of the white calibration panel at the beginning and end of each series. Vegetation cover was measured from one corner of the site to the opposite corner in accordance with the procedure for conducting field inspections of plants for the presence of pathogens. The area of one measurement covered by the spectroradiometer sensor was 0.222 m^2^. This can be considered as the spatial resolution of ground measurements. Thus, the area of measurements of each test plot of the main experiment was about 7 m^2^, and for small plots of the experiment, it was 2.22 m^2^. The total measurement area of all test plots was about 60 m^2^.

Data obtained from ground-based spectrometric measurements are a set of spectral brightness coefficient (SBC) values that indicate the degree to which sunlight is reflected from plant surfaces at each wavelength. These data were processed automatically using a specially written script in the Python programming language.

### 4.4. UAV Imagery

For aerial photography, a Parrot SEQUOIA+ multispectral camera was used [52]. It allows multi-zone imaging in four channels with central spectral values of 550 nm, 660 nm, 735 nm, and 790 nm. As a result, a series of images of the experiment from the air was created, interconnected into a single orthomosaic using the Pix4D software 4.8.0, 4.8.1, 4.8.2. A radiometric calibration target was used to calibrate the images, which allows one to calibrate and correct the reflectivity of the images in accordance with the specified values. Three images were taken using the Parrot Sequoia camera. The calculated spatial resolution for them was about 7 cm, being 7.0 cm for the 19 May survey, 7.1 cm for the 27 May survey, and 7.5 cm for the 3 June 2022 survey. The data density obtained from the UAV was 27.8 Mb/ha for one image or 83.5 Mb/ha for the series of three images used in the study.

### 4.5. Satellite Imagery Data

For the test plots, ultra-high resolution satellite imagery data were obtained by the private space company Planet (San Francisco, CA, USA) using the Dove Planet satellite constellation. The spatial resolution of the images is 50 cm. The spectral resolution is 8 channels with central spectral values of 443 nm, 490 nm, 531 nm, 565 nm, 610 nm, 665 nm, 705 nm, and 865 nm. The amount of data used in the study was calculated to be 57 Kb/ha for a single satellite image, or 0.92 MB/ha for the series of 16 satellite images used in the study.

Unfortunately, both the area of the small-plot experiment and the edge effect influence prevented obtaining valid data. Therefore, the satellite data analysis for the small experiment was not carried out.

### 4.6. Data Processing

Spectral ranges of the Parrot SEQUOIA+ camera as well as the NDVI vegetation index were to ensure the comparability of spectrometric data obtained using different imaging systems. Aerial and satellite imagery data were processed using the Zonal Statistic tool provided by the open-source desktop geoinformation system QGIS. As a result, the spectral data for each area were recorded in the form of a general table as the arithmetic mean of all pixels that fell within the boundaries of the test areas in the image.

When extracting spectral brightness values, edge pixels were excluded to avoid mixed pixels. The test sites were marked on the ground with the help of special markers. This allowed the angles of the sites to be determined. Spectra inside these sites were taken with a 1 m indentation inside the boundaries.

Correlation analysis of ground-based spectrometry and aerial survey data for 27–28 May (the period where both types of surveys were carried out) showed the following: despite the different densities of repeated measurements within the main plots and areas of the small-plot experiment (0.3 meas./m^2^ versus 5 meas./m^2^), a stable correlation of the obtained values can be traced only for the red and IR regions of the spectrum (Table 11). This suggests that differences in the pathogenic background can be reliably detected using UAVs only in these ranges. This also suggests that it is not necessary to strive for a high density of measurements per unit area in order to conduct ground-based spectrometry.

Correlation analysis of the relationship between spectral data and disease records was carried out using the SciPy library of the Python programming language (https://scipy.org/ accessed on 2 August 2023). Correlation analysis of the relationship between the disease development and air pollution indicator was carried out on the basis of non-parametric statistics methods using the Spearman test at a high 95% significance level using the Statistica 2010 program.

The impact of disease development on the spectral characteristics of winter wheat crops in different regions of the spectrum was assessed using the methods of single-factor analysis of variance. For data analysis, individual channels of the spectrum were selected from the total operating range of the spectroradiometer: 490 nm, 550 nm, 660 nm, 720 nm, 845 nm, 1445 nm, 1675 nm, and 2345 nm. These spectral ranges are widely used in the study of plants and are closely related to their biophysical characteristics. Statistical data processing in the selected regions of the spectrum was carried out with the calculation of the mean value and standard deviation. During the analysis, the variable values of the coefficient of spectral brightness were grouped into categories corresponding to different degrees of disease development. Correlation analysis of disease development with the indicator of the number of detected pathogen spores was carried out on the basis of nonparametric statistics methods using the Spearman correlation coefficient at a high significance level of 95%.

All of the above statistical methods of analysis were performed in the Statistica 2010 program.

## 5. Conclusions

We applied various methods for obtaining spectral information on winter wheat crops and compared the obtained data with the level of pathogenic background of the test plots. As a result, it was found that the least disease-prone plants predominantly had lower values in the green, red, and red-edge ranges of the spectrum and high SBC values in the IR range during the development phase “flag list”–“wax ripeness”.

Red and IR ranges are the most informative spectral ranges when using satellite images and multispectral cameras placed on UAVs. In this case, the high frequency of measurements is more important for determining the level of the pathogenic background than the spectral resolution. It was also determined that when carrying out ground-based spectrometry, information acquisition density does not play as significant of a role as the repetition of measurements. The use of vegetation indices in assessing the dynamics of the spectral image of various survey systems allows us to bring them to similar values.

## Figures and Tables

**Figure 1 plants-12-03223-f001:**
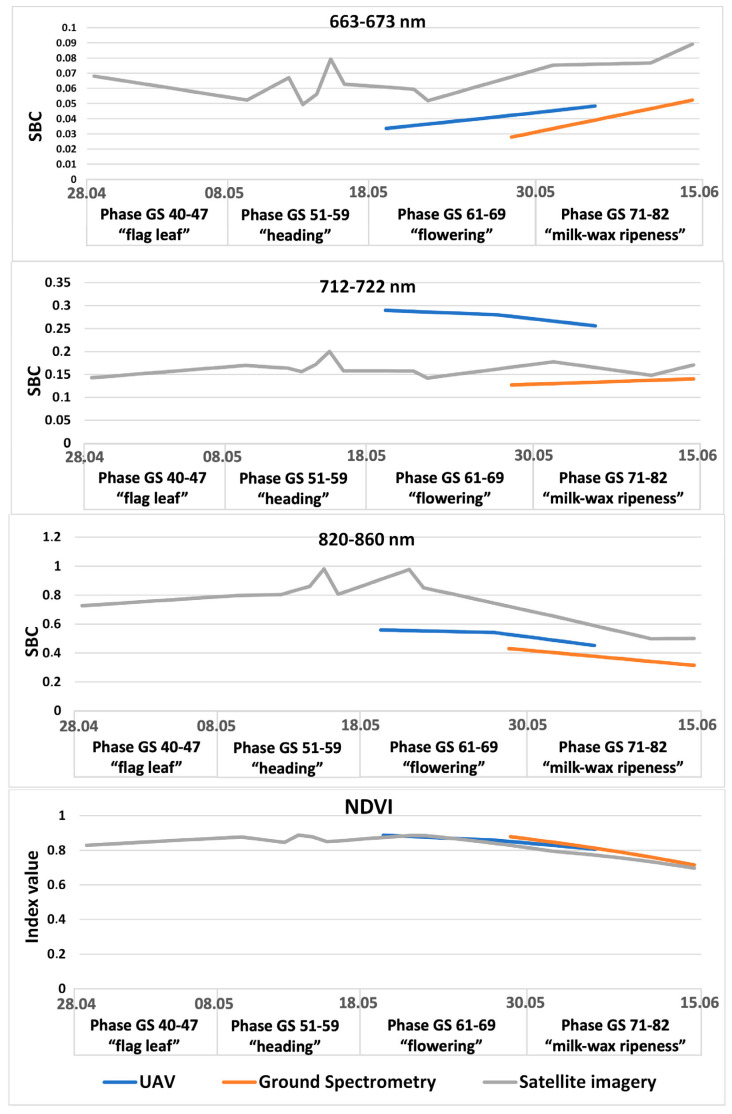
Dynamics of development of pathogens and SBCs in different spectral ranges, from different sources.

**Figure 2 plants-12-03223-f002:**
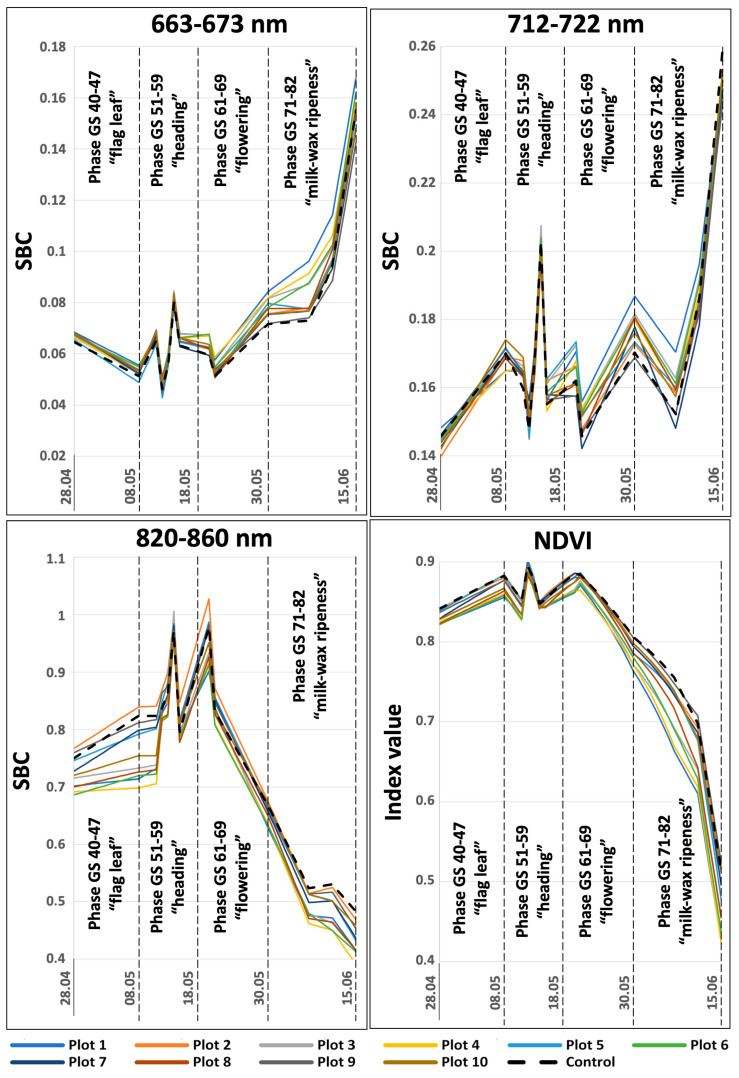
SBC dynamics in different spectral ranges in individual test plots, obtained from satellite imagery data.

**Figure 3 plants-12-03223-f003:**
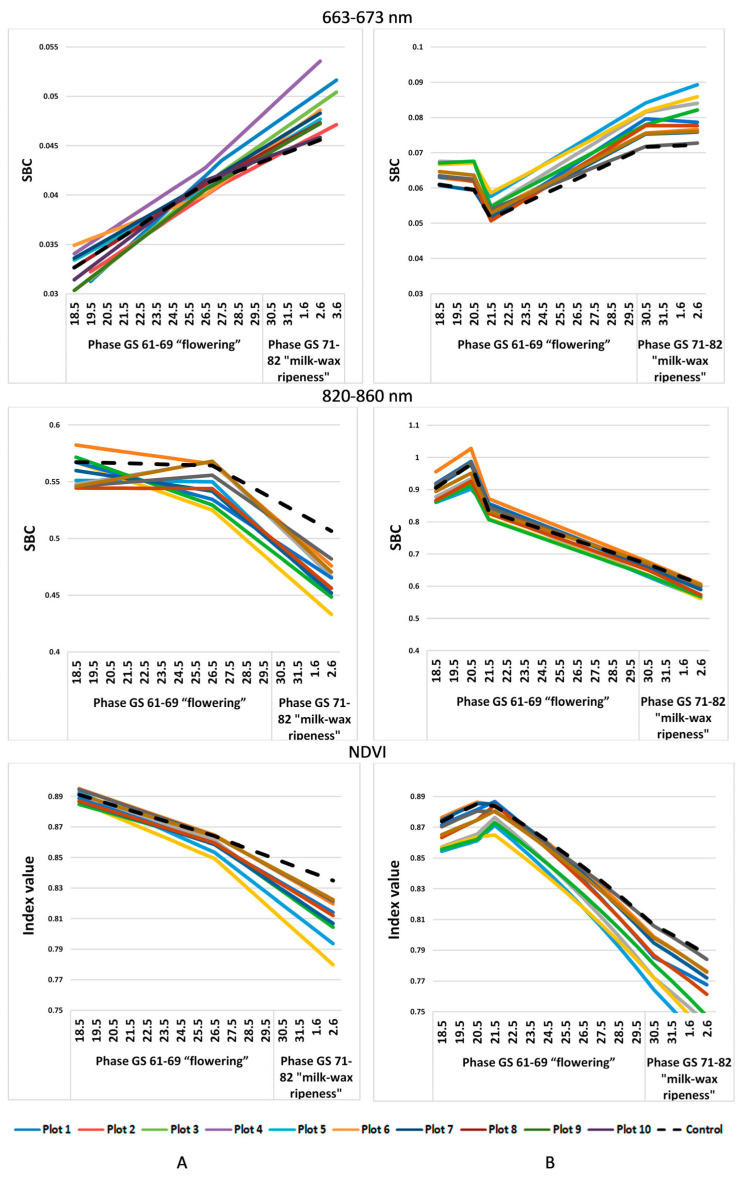
SBC dynamics in different spectral ranges in individual test plots, obtained from the data of (**A**) satellite imagery; (**B**) UAVs.

**Figure 4 plants-12-03223-f004:**
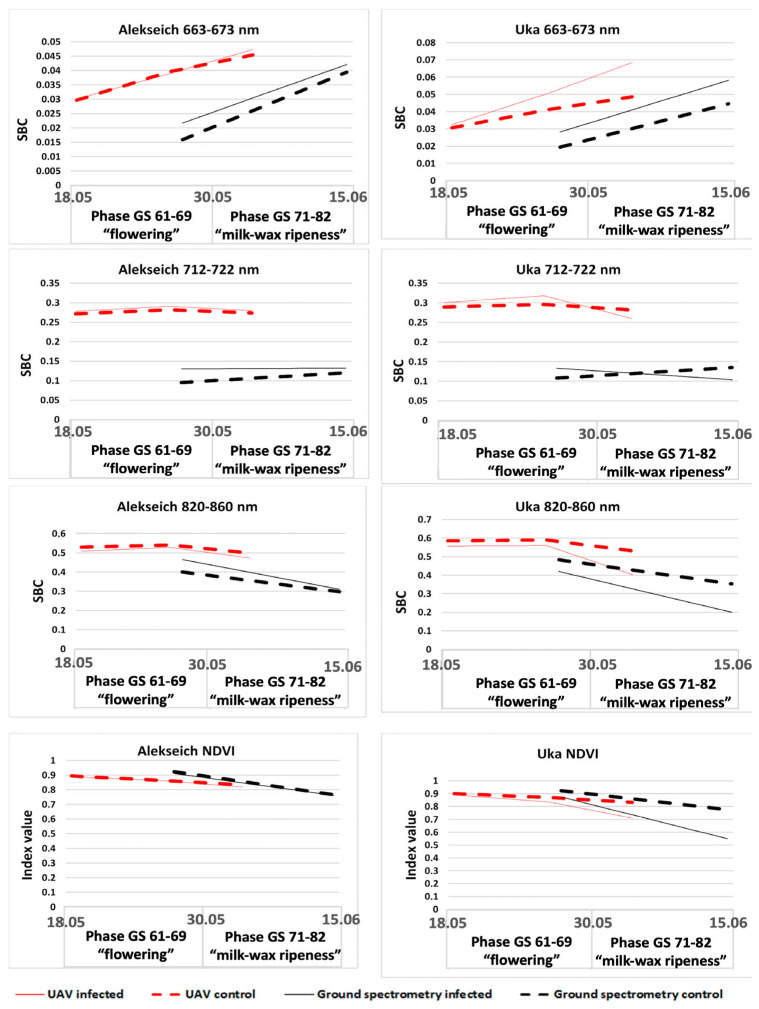
SBC dynamics in different spectral ranges in a small-scale experiment with crops of Alexech and Yuka varieties, obtained from the data of ground-based spectrometric measurements and UAV surveys.

**Figure 5 plants-12-03223-f005:**
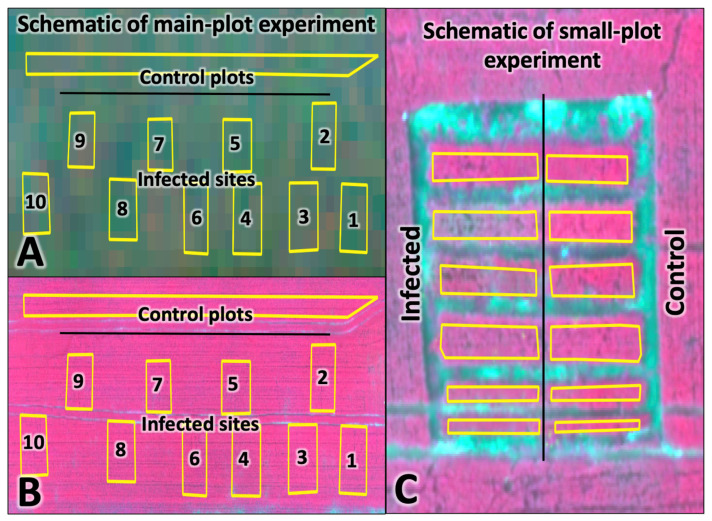
Remote sensing data obtained for the experimental site: (**A**) Dove Planet (26 May 2022); (**B**,**C**) Parrot SEQUOIA+ (27 May 2022). The contours show the areas of the experiment.

**Table 1 plants-12-03223-t001:** The results of the correlation analysis of the dependence between the degree of development of diseases in the winter wheat crops of the main test plots and the values of the SBC of the spectral channels of the Dove Planet satellite constellation.

Pathogen	Spectral Channels								Vegetation Indices		
	443	490	531	565	610	665	705	865	NDVI	NDVI2	GNDVI
GS 40−47 “flag leaf”											
Powdery	−0.49	0.75 *	0.05	−0.67 *	0.21	0.21	−0.42	−0.09	−0.16	−0.63 *	−0.57
mildew											
Septoria	−0.49	−0.17	−0.35	−0.28	0.59	−0.38	0.11	−0.37	−0.12	0.53	0.16
Yellow rust	−0.43	0.24	−0.27	−0.62	0.59	−0.18	−0.16	−0.01	0.07	0.05	−0.21
Yellow spot	-	-	-	-	-	-	-	-	-	-	-
Brown rust	-	-	-	-	-	-	-	-	-	-	-
GS 50−59 “heading”											
Powdery	0.74	−0.43	0.14	0.06	0.48	−0.04	−0.24	−0.40	−0.33	0.12	−0.06
mildew											
Septoria	0.67	0.79 *	0.13	−0.59	−0.17	−0.50	−0.18	0.06	0.49	0.39	0.17
Yellow rust	0.69 *	0.28	0.46	−0.10	−0.25	0.08	−0.05	−0.04	0.15	0.12	0.02
Yellow spot	−0.79 *	−0.45	−0.03	−0.18	0.72 *	0.10	0.05	−0.30	−0.32	−0.20	−0.21
Brown rust	0.42	−0.16	−0.13	0.36	0.28	0.30	0.24	−0.19	−0.30	−0.34	−0.29
GS 60−70 “flowering”											
Powdery	−0.04	−0.23	−0.07	−0.27	0.04	0.09	0.05	−0.27	−0.18	−0.08	−0.24
mildew											
Septoria	0.40	−0.21	−0.18	0.00	−0.23	0.01	−0.16	−0.03	−0.01	−0.24	0.06
Yellow rust	0.76 *	0.05	−0.14	0.33	0.14	0.22	0.01	−0.28	−0.26	−0.39	−0.19
Yellow spot	0.12	0.38	0.03	−0.04	−0.03	−0.31	−0.06	0.64 *	0.46	0.49	−0.22
Brown rust	-	-	-	-	-	-	-	-	-	-	-
GS 71−82 “milk−wax ripeness”											
Powdery	−0.10	−0.19	−0.34	−0.09	0.12	−0.11	−0.15	0.59	0.31	0.00	0.11
mildew											
Septoria	−0.02	−0.18	0.17	−0.35	0.23	−0.43	0.13	0.23	0.44	0.73 *	0.45
Yellow rust	−0.34	0.11	−0.15	0.01	0.53	−0.04	0.28	0.42	0.19	0.33	0.06
Yellow spot	0.10	0.29	0.48	0.42	0.35	0.28	0.00	−0.23	−0.33	−0.39	−0.18
Brown rust	0.65 *	−0.39	−0.11	−0.15	−0.21	−0.33	−0.56	0.25	0.36	−0.08	0.42

Notes: * Statistical significance of data correlation is confirmed.

**Table 2 plants-12-03223-t002:** The results of the correlation analysis of dependence between the degree of disease development in winter wheat crops of the main sample areas and the values of SBC spectral channels of the Parrot SEQUOIA+ multispectral camera are presented.

Pathogen	Spectral Channels				Vegetation Indices		
	550	660	735	790	NDVI	NDVI2	GNDVI
GS 50–59 “heading”							
Powdery	0.73 *	0.68 *	0.44	0.20	−0.63	−0.72 *	0.27
mildew							
Septoria	−0.55	−0.66 *	−0.17	−0.28	0.55	0.66 *	−0.28
Yellow rust	−0.28	−0.15	−0.36	0.00	0.12	0.10	−0.67 *
Yellow spot	0.22	0.07	0.09	−0.22	−0.16	−0.32	0.31
Brown rust	0.17	0.17	0.00	0.09	−0.17	0.00	−0.17
GS 60−70 “flowering”							
Powdery	−0.23	0.22	−0.31	0.50	−0.44	−0.81 *	−0.55
mildew							
Septoria	−0.02	−0.20	−0.26	−0.47	0.22	0.56	0.72 *
Yellow rust	−0.64 *	−0.33	−0.33	0.34	−0.11	−0.10	0.03
Yellow spot	0.12	−0.12	0.15	−0.14	0.39	−0.18	−0.08
Brown rust	-	-	-	-	-	-	-
GS 71−82 “milk−wax ripeness”							
Powdery	−0.43	−0.24	−0.50	0.38	0.31	0.08	0.03
mildew							
Septoria	0.11	−0.12	0.41	0.25	0.19	0.23	0.25
Yellow rust	−0.04	−0.02	−0.18	0.23	0.12	−0.03	0.01
Yellow spot	0.43	0.60	−0.16	−0.45	−0.58	−0.61	−0.59
Brown rust	−0.35	−0.08	−0.29	−0.18	−0.02	−0.01	−0.18

Notes: * Statistical significance of data correlation is confirmed.

**Table 3 plants-12-03223-t003:** The results of the correlation analysis of the relationship between the number of spores caught using the PSL-3 device and the degree of disease development for three growing seasons during the study period.

Pathogen	Correlation Coefficient	Level of Statistical Significance
Phase GS 40–47 “flag leaf”		
Powdery mildew	0.7	0.053
Yellow spot	−0.1	0.984
Septoria	-	-
Yellow rust	0.8 *	0.004 *
Phase GS 61 “early flowering”		
Powdery mildew	0.7 *	0.040
Yellow spot	-	-
Septoria	-	-
Yellow rust	0.8 *	0.004
Phase GS 71–82 “milk−wax ripeness”		
Powdery mildew	0.7 *	0.008 *
Yellow spot	0.3	0.339
Septoria	-	-
Yellow rust	0.9 *	0.001 *

Notes: R is the correlation coefficient index; *p* is the level of statistical significance (*p* ˂ 0.05); * statistical significance of data correlation is confirmed.

**Table 4 plants-12-03223-t004:** Assessment of the influence of the disease development factor on the spectral characteristics of winter wheat crops in different ranges of the spectrum (GS 60–70 “flowering”).

Spectral Range, nm	Powdery Mildew		Septoria		Yellow Rust		Generalized Categories	
	F	*p*	F	*p*	F	*p*	F	*p*
490	17.3 *	0.00 *	7.2 *	0.00 *	6.4 *	0.00 *	12.6 *	0.00 *
550	13.4 *	0.00 *	5.9 *	0.00 *	5.3 *	0.00 *	9.8 *	0.00 *
660	16.3 *	0.00 *	7.9 *	0.00 *	5.8 *	0.00 *	13.7 *	0.00 *
720	8.5 *	0.00 *	2.6 *	0.03 *	2.1	0.07	6.6 *	0.00 *
845	3.9 *	0.04 *	1.2	0.33	1.9	0.09	4.3 *	0.00 *
1445	7.1 *	0.00 *	2.5 *	0.04 *	3.2 *	0.00 *	5.1 *	0.00 *
1675	8094.3 *	0.00 *	0.9	0.510	1.5	0.202	3.5 *	0.00 *
2005	6.2 *	0.00 *	1.6	0.160	3.1 *	0.01 *	4.1 *	0.00 *
2345	5.4 *	0.00 *	1.6	0.146	2.9 *	0.01 *	3.8 *	0.00 *

Notes: * A mathematically reliable influence of the disease development factor on the value of the spectral brightness coefficient is found.

**Table 5 plants-12-03223-t005:** Results of a posteriori analysis of the spectral characteristics of winter wheat crops with different gradations of development of powdery mildew according to the Duncan criterion (GS 60–70 “flowering”). Main plot experiments.

R, %	Spectral Range, nm								
	490	550	660	720	845	1445	1675	2005	2345
1.5	0.0245 ±0.0006 b	0.0563 ±0.0011 b	0.0295 ±0.0007 b	0.149 ±0.0023 b	0.469 ±0.0070 c	0.0539 ±0.0012 bc	0.145 ±0.0027 b	0.0260 ±0.0008 b	0.0376 ±0.0009 b
2.5	0.0217 ±0.0007 a	0.0514 ±0.0013 a	0.0259 ±0.0009 a	0.139 ±0.0028 a	0.444 ±0.0087 ab	0.0507 ±0.0015 ab	0.135 ±0.0034 ab	0.0240 ±0.0009 ab	0.0339 ±0.0011 a
3.5–4	0.0219 ±0.0004 a	0.0514 ±0.0009 a	0.0268 ±0.0006 a	0.135 ±0.0020 a	0.440 ±0.0054 a	0.0481 ±0.0010 a	0.133 ±0.0021 a	0.0223 ±0.0006 a	0.0336 ±0.0007 a
Control	0.0272 ±0.0006 c	0.0608 ±0.0010 c	0.0330 ±0.0009 c	0.151 ±0.0031 b	0.464 ±0.0096 bc	0.0568 ±0.0016 c	0.143 ±0.0037 bc	0.0266 ±0.0011 b	0.0383 ±0.0013 b

Notes: R—an indicator of the degree of progression of the disease; data represent the average mean value of the SBC and standard error. In a column, the average values with the same letter do not differ significantly.

**Table 6 plants-12-03223-t006:** Results of a posteriori analysis of the spectral characteristics of winter wheat crops with different gradations of development of septoria according to the Duncan criterion (GS 60–70 “flowering”). Main plot experiments.

R, %	Spectral Range, nm								
	490	550	660	720	845	1445	1675	2005	2345
1–1.5	0.0248 ±0.0006 a	0.0523 ±0.0009 a	0.0274 ±0.0006 a	0.141 ±0.0021 a	0.456 ±0.0062 ab	0.0512 ±0.0011 a	0.139 ±0.0024 a	0.0239 ±0.0007 ab	0.0354 ±0.0008 ab
2	0.0246 ±0.0005 a	0.0531 ±0.0008 a	0.0275 ±0.0006 a	0.142 ±0.0019 a	0.446 ±0.0056 ab	0.0502 ±0.0009 a	0.135 ±0.0022 a	0.0235 ±0.0006 a	0.0342 ±0.0007 a
2.5	0.0244 ±0.0007 a	0.0523 ±0.0019 a	0.0268 ±0.0013 a	0.139 ±0.0042 a	0.437 ±0.0012 a	0.0524 ±0.0021 a	0.139 ±0.0048 a	0.0248 ±0.0014 ab	0.0354 ±0.0016 ab
Control	0.0297 ±0.0009 b	0.0608 ±0.0015 b	0.0330 ±0.0010 b	0.151 ±0.0033 b	0.464 ±0.0098 b	0.0568 ±0.0016 b	0.143 ±0.0038 a	0.0266 ±0.0011 b	0.0382 ±0.0013 b

Notes: R—an indicator of the degree of progression of the disease; data represent the average mean value of the SBC and standard error. In a column, the average values with the same letter do not differ significantly.

**Table 7 plants-12-03223-t007:** Results of a posteriori analysis of the spectral characteristics of winter wheat crops with different gradations of development of yellow rust according to the Duncan criterion (GS 60–70 “flowering”). Main plot experiments.

R, %	Spectral Range, nm								
	490	550	660	720	845	1445	1675	2005	2345
2.5–3.5	0.0252 ±0.0008 a	0.0543 ±0.0014b	0.0279 ±0.0009 a	0.145 ±0.0029 ab	0.460 ±0.0087 a	0.0539 ±0.0012 a	0.140 ±0.0034 ab	0.0247 ±0.0010 bc	0.0358 ±0.0011 ab
4.0–4.5	0.0249 ±0.0008 a	0.0523 ±0.0013 a	0.0276 ±0.0009 a	0.140 ±0.0029 a	0.464 ±0.0087 a	0.0507 ±0.0015 a	0.139 ±0.0034 ab	0.0229 ±0.0010 ab	0.0347 ±0.0011 ab
5.5–6.0	0.0250 ±0.0007 a	0.0498 ±0.0009 a	0.0280 ±0.0007 a	0.142 ±0.0024 ab	0.437 ±0.0072 a	0.0481 ±0.0010 ab	0.138 ±0.0028 ab	0.0248 ±0.0008 bc	0.0361 ±0.0009 b
8.0–10.0	0.0232 ±0.0008 a	0.0573 ±0.0018 bc	0.0261 ±0.0009 a	0.139 ±0.0030 a	0.449 ±0.0088 a	0.0549 ±0.0020 a	0.130 ±0.0034 a	0.0214 ±0.0010 a	0.0320 ±0.0012 a
15.0	0.0244 ±0.0011 a	0.0573 ±0.0018 bc	0.0268 ±0.0013 a	0.139 ±0.0041 a	0.437 ±0.0123 a	0.0549 ±0.0020 ab	0.140 ±0.0047 ab	0.0249 ±0.0014 bc	0.0355 ±0.0016 ab
Control	0.0297 ±0.0008 b	0.0608 ±0.0010 c	0.0330 ±0.0009 b	0.151 ±0.0031 b	0.464 ±0.0096 a	0.0568 ±0.0016 b	0.143 ±0.0037 b	0.0266 ±0.0011 c	0.0382 ±0.0013 b

Notes: R—an indicator of the degree of progression of the disease; data represent the average mean value of the SBC and standard error. In a column, the average values with the same letter do not differ significantly.

**Table 8 plants-12-03223-t008:** Measurement dates of test plots (main and small-plot experiments).

Type of Research	Phase	Time Period
Field inspections Satellite imagery	GS 40–47 “flag leaf”	27 April–5 May 2022
Field inspections Ground spectrometry UAV Satellite imagery	Z 50–70 “heading—early flowering”	8 May–30 May 2022
Field inspections UAV	Z 71–82 “milk-wax ripeness”	31 May–15 June 2022

**Table 9 plants-12-03223-t009:** Average indicators of disease development in winter wheat crops of the main test plots based on the results of field surveys.

Pathogen	Plots										Control
	1	2	3	4	5	6	7	8	9	10	
GS 40–47 “flag leaf 28 April 2022											
Powdery mildew	1.27	1.07	0.47	1	1.07	2.93	0.87	2.4	1	4	0.4
Septoria	0	0	0	0.67	0	0	0	0	0	0	0
Yellow rust	0	0.4	0	6.67	0	0	0	0	1.33	3.67	0
Yellow spot	0	0	0	0	0	0	0	0	0	0	0
Brown rust	0	0	0	0	0	0	0	0	0	0	0
GS 50–59 “heading” 18 May 2022											
Powdery mildew	0.68	2.01	0.83	3.29	1.32	2.14	1.84	2.98	0.76	2.02	0
Septoria	1.36	1.16	1.38	1.06	1.48	0.48	0.92	0.71	2.34	1.79	0.05
Yellow rust	4.08	3.59	6.70	3.67	1.97	1.42	4.78	1.68	1.68	8.31	0.08
Yellow spot	0.01	0	0	0.06	0.06	0	0	0.06	0.03	0	0
Brown rust	0	0	0.01	0.01	0	0	0	0	0	0	0
GS 60–70 “flowering” 25 May 2022											
Powdery mildew	4.56	3.27	2.48	3.84	1.38	1.40	3.86	3.86	1.37	2.40	0.62
Septoria	1.09	0.92	2.11	1.79	1.69	1.34	1.73	1.38	1.98	2.44	1.43
Yellow rust	4.28	4.44	7.89	10.12	2.81	6.28	5.89	6.17	3.36	16.89	0.09
Yellow spot	0	0	0	0	0	0	0	0	0	0	0
Brown rust	0	0	0	0	0	0	0	0	0	0	0
GS 71–82 “milk-wax ripeness” 2 June 2022											
Powdery mildew	0	0.33	0	0	0	0	0	0	0	0	0
Septoria	3.94	8.14	12.22	3.78	11.20	4.50	5.39	8.33	2.92	4.17	1.17
Yellow rust	12.39	16.17	11.44	7.94	11.11	5.22	5.61	11.06	4.83	9.78	0.6
Yellow spot	0	0.09	0.22	0.31	0.06	0	0.22	0.03	0	0	0
Brown rust	0	0	0	0	0.17	0	4.61	0	0	0	0

**Table 10 plants-12-03223-t010:** Indicators of the average development of diseases, chlorophyll content, and plant height of winter wheat crops in a small-plot experiment.

Variety	Powdery Mildew	Septoria	Yellow Rust
GS 50–59 “heading” 18 May 2022			
Alekseich control	0.01	1.73	0.72
Alekseich infectious	0	0.8	12.39
Yuka control	0	10	0.74
Yuka infectious	0.09	5.23	0.34
GS 60–70 “flowering” 25 May 2022			
Alekseich control	0.22	0.06	1.17
Alekseich infectious	0	0.89	41.72
Yuka control	0.62	1.43	0.09
Yuka infectious	0	2.03	0.29
GS 71–82 “milk-wax ripeness” 2 June 2022			
Alekseich control	0.24	7.76	2.54
Alekseich infectious	0	1	38.8
Yuka control	0.31	1.17	0.6
Yuka infectious	0	3.21	1.06

**Table 11 plants-12-03223-t011:** SBC correlation coefficients of ground-based spectrometry data and aerial survey data of test plots.

Test Plots	Spectral Range, nm			
	550–570	663–673	712–722	820–860
Main plots	0.23	0.70	0.38	0.85
Small-plot experiment	−0.12	0.82	−0.22	0.63

## Data Availability

Not applicable.

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
