# Peer review of "Development of Methods for Remote Monitoring of Leaf Diseases in Wheat Agrocenoses"

_plants, 2023, doi:10.3390/plants12183223_

Round 1
Reviewer 1 Report
Major comments
Authors write that "Ten test plots were allocated to create an artificial infectious background and, accordingly, 10 control plots to ensure comparability of aerospace survey data with the results of ground-based spectrometric measurements within the experimental field with a total area of 1 ha. The size of each test plot was 10 × 10 m (100 m2)."
This is unclear: 10 test plots + 10 control plots = 20 plots x 100 m2= 2000 m2. What about the remaining area (8000 m2)?
Why only one mapping from UAV?
Please add ground resolution for sequoia imaging
Please add ground resolution/total area for ground measurment
Please add more information regarding calibration of images collected by UAV/Sequoia
Please add a figure of UAV and one of satellite of the study area
Satellite data have a ground resolution of 0.5 m: how were plots borders considered? were they excluded from analyses? Please provide more information on data processing.
The approach reported in the paper is apparently using a relevant amount of data: thus it would be interesting to add the “digitization footprint” (Mb/ha or Gb/ha) information
Please provide more information on how the actual infeection of plots have been evaluated.
Minor comments
Remove quotation marks from the title
Improve readability of fonts in the graphs/figures (increase font size, without increasing the total dimension of graphs)
"Space" or "aerospace imaging" terms are not commonly used in agriculture: "remote sensing" or "satellite" is much more common
Avoid using "we": preferred using impersonal form
english is sufficient
Author Response
Responses to reviewer №1
We sincerely thank reviewer for constructive and useful comments and have tried to address all of them. The corrections have been highlighted in yellow in the text of the paper for the ease of reference.
Our detailed responses are below.
Reviewer #1:
Major comments
Authors write that "Ten test plots were allocated to create an artificial infectious background and, accordingly, 10 control plots to ensure comparability of aerospace survey data with the results of ground-based spectrometric measurements within the experimental field with a total area of 1 ha. The size of each test plot was 10 × 10 m (100 m2)."
This is unclear: 10 test plots + 10 control plots = 20 plots x 100 m2= 2000 m2. What about the remaining area (8000 m2)?
Why only one mapping from UAV?
Please add ground resolution for sequoia imaging
Please add ground resolution/total area for ground measurment
Please add more information regarding calibration of images collected by UAV/Sequoia
Please add a figure of UAV and one of satellite of the study area
Satellite data have a ground resolution of 0.5 m: how were plots borders considered? were they excluded from analyses? Please provide more information on data processing.
The approach reported in the paper is apparently using a relevant amount of data: thus it would be interesting to add the “digitization footprint” (Mb/ha or Gb/ha) information
Please provide more information on how the actual infeection of plots have been evaluated.
Minor comments
Remove quotation marks from the title
Improve readability of fonts in the graphs/figures (increase font size, without increasing the total dimension of graphs)
"Space" or "aerospace imaging" terms are not commonly used in agriculture: "remote sensing" or "satellite" is much more common
Avoid using "we": preferred using impersonal form
Response: All comments and suggestions were accepted, corrected and additions were made to the article.

Reviewer 2 Report
General comments
The major recommendations I have are 1) Please put the methods after the introduction and make sure you explain not only the data but also your analysis methods, which include the processing of the satellite and other datasets and the correlation approaches. Please use a consistent label for each dataset used. Please note that ‘space’ is not a relevant label and is actually inaccurate – use the actual dataset name defined in the methods section (Planet imagery, for example). 2) Refocus the analysis on the evolution of the fungal disease in your test vs control plots. Your point in the introduction that spectral analysis could predict two weeks before an outbreak should be the focus. The relationship between the different spectral datasets is highly uninteresting from my perspective – we’ve known these correlations for forty years – literally from 1972. What is SUPER interesting and novel is the correlation between the spectral observations and the development of the fungal disease in the crop. Please focus on that. 3) this paper would be far more interesting with more agronomic insights – what is the value of early detection? How will it change wheat management in Russia?
Specific comments
Lines 54-56 - what does it mean ‘environmental factors’ on ‘data acquisition’ – do you mean acquisition of satellite data? If so, what environmental factors do you mean? Also for ‘transformation of models’ transferred to a ‘new index system’ – do you mean when you are applying the model to a new region? If so please say that. Otherwise, please define ‘new index system’ – what new index?
Lines 61-62 – by cultivar, I assume you mean different varieties of wheat – please state that. Cultivar means several different things and can be ambiguous in a diverse disciplinary community.
Line 71 – time series of data of what? Disease prevalence? Or remote sensing time series?
Please put the Materials and Methods section before the Results section – not sure why it is out of order here.
Table 3 – this is confusing – I think it would be better to put Field inspections in the column on the left and the other data acquisitions on the right.
Table 3 – by ‘space photography’ do you mean the ‘ground-based spectrometry’ mentioned in the next paragraph, the ‘aerial photography’ discussed on line 319, or the Planet labs observations mentioned in line 323? Please be explicit here – tell us EXACTLY what data you are talking about.
Now, back to the results section!
Line 98, Figure 1 – how is the ‘average progression of the disease, %’ calculated? It does not seem to be presented in the methods section. Please provide that – is it from the standard protocols you mentioned 283-284? If so, please state exactly how you calculated the percentage, as the ‘standard protocols’ are quite unknown to the remote sensing community, who should be interested in citing this paper if it is clear.
Lines 113-120 – I think these are methods and should be put in a methods section, explaining how the correlations were done. Also, please explain the ‘aerial survey’ in the methods– is this the sampling of spores in the air above the crop that was mentioned in the introduction?
Line 122 – what is SBC? Acronym does not appear to be defined.
Table 1 – what are the two things being correlated here? Why don’t you present the ground-based infection rates instead of spectrometry data? It is hardly an interesting conclusion that the satellite data is able to capture the same information as the spectrometer on the ground – this has been shown hundreds of times in the past forty years in the literature. I would just delete this unless you can present information regarding the fungal infection instead of merely reflectance.
Figure 2 – what are the numbers on the X axis? Dates? Please provide a label of the axis on this plot. Also, again the SBC label instead of ‘index value’ for the different spectral indices. I assume that for the first three figures is spectral reflectance, calculated from the Planet imagery, but it is not clear. Please revise the caption of this figure to specify ‘Planet imagery’ and define SBC.
Lines 247-250 – interesting conclusion that very high resolution (VHR) imagery would be ‘less expensive’ than ground spectrometry. There are a lot of additional costs for acquiring, processing, and delivering VHR imagery beyond simply purchasing it, which include having access to technical expertise. Also please explain what you mean by ‘less expensive analogues of spectrometric studies.’ – This I don’t get. What studies?
Line 263 - Additional discussion points that could be added here include how predictive the authors think this system is. Can the satellite data identify an emerging problem sufficiently early so that the farmer could intervene in the development of the fungal pathogen before yield impacts occur? This is the central question that the paper should address.
This has the potential to be an excellent paper. Unfortunately it now suffers from a wide variety of language, presentation and organizational problems that need to be addressed before it can be published. The paper *seems* like it is in English, but it is more likely to have been written with a translator or some sort of bot. I really recommend that the authors find an actual English speaker in their field to review the paper before resubmission. I have pointed out a few of the most confusing parts, but nearly every paragraph is a bit wonky. Hard to understand the paper in general, but it does seem that the overall analysis has great promise. Please revise and resubmit this paper!
Author Response
Responses to reviewer №2
We sincerely thank both reviewers for constructive and useful comments and have tried to address all of them. The corrections have been highlighted in yellow in the text of the paper for the ease of reference.
Our detailed responses are below.
Reviewer #2:
General comments
The major recommendations I have are 1) Please put the methods after the introduction and make sure you explain not only the data but also your analysis methods, which include the processing of the satellite and other datasets and the correlation approaches. Please use a consistent label for each dataset used. Please note that ‘space’ is not a relevant label and is actually inaccurate – use the actual dataset name defined in the methods section (Planet imagery, for example). 2) Refocus the analysis on the evolution of the fungal disease in your test vs control plots. Your point in the introduction that spectral analysis could predict two weeks before an outbreak should be the focus. The relationship between the different spectral datasets is highly uninteresting from my perspective – we’ve known these correlations for forty years – literally from 1972. What is SUPER interesting and novel is the correlation between the spectral observations and the development of the fungal disease in the crop. Please focus on that. 3) this paper would be far more interesting with more agronomic insights – what is the value of early detection? How will it change wheat management in Russia?
Response: All comments and suggestions were accepted, corrected and additions were made to the article.
Specific comments
Lines 54-56 - what does it mean ‘environmental factors’ on ‘data acquisition’ – do you mean acquisition of satellite data? If so, what environmental factors do you mean? Also for ‘transformation of models’ transferred to a ‘new index system’ – do you mean when you are applying the model to a new region? If so please say that. Otherwise, please define ‘new index system’ – what new index?
Lines 61-62 – by cultivar, I assume you mean different varieties of wheat – please state that. Cultivar means several different things and can be ambiguous in a diverse disciplinary community.
Line 71 – time series of data of what? Disease prevalence? Or remote sensing time series?
Please put the Materials and Methods section before the Results section – not sure why it is out of order here.
Table 3 – this is confusing – I think it would be better to put Field inspections in the column on the left and the other data acquisitions on the right.
Table 3 – by ‘space photography’ do you mean the ‘ground-based spectrometry’ mentioned in the next paragraph, the ‘aerial photography’ discussed on line 319, or the Planet labs observations mentioned in line 323? Please be explicit here – tell us EXACTLY what data you are talking about.
Line 98, Figure 1 – how is the ‘average progression of the disease, %’ calculated? It does not seem to be presented in the methods section. Please provide that – is it from the standard protocols you mentioned 283-284? If so, please state exactly how you calculated the percentage, as the ‘standard protocols’ are quite unknown to the remote sensing community, who should be interested in citing this paper if it is clear.
Lines 113-120 – I think these are methods and should be put in a methods section, explaining how the correlations were done. Also, please explain the ‘aerial survey’ in the methods– is this the sampling of spores in the air above the crop that was mentioned in the introduction?
Line 122 – what is SBC? Acronym does not appear to be defined.
Table 1 – what are the two things being correlated here? Why don’t you present the ground-based infection rates instead of spectrometry data? It is hardly an interesting conclusion that the satellite data is able to capture the same information as the spectrometer on the ground – this has been shown hundreds of times in the past forty years in the literature. I would just delete this unless you can present information regarding the fungal infection instead of merely reflectance.
Figure 2 – what are the numbers on the X axis? Dates? Please provide a label of the axis on this plot. Also, again the SBC label instead of ‘index value’ for the different spectral indices. I assume that for the first three figures is spectral reflectance, calculated from the Planet imagery, but it is not clear. Please revise the caption of this figure to specify ‘Planet imagery’ and define SBC.
Lines 247-250 – interesting conclusion that very high resolution (VHR) imagery would be ‘less expensive’ than ground spectrometry. There are a lot of additional costs for acquiring, processing, and delivering VHR imagery beyond simply purchasing it, which include having access to technical expertise. Also please explain what you mean by ‘less expensive analogues of spectrometric studies.’ – This I don’t get. What studies?
Line 263 - Additional discussion points that could be added here include how predictive the authors think this system is. Can the satellite data identify an emerging problem sufficiently early so that the farmer could intervene in the development of the fungal pathogen before yield impacts occur? This is the central question that the paper should address.
Response: All comments and suggestions were accepted, corrected and additions were made to the article.

Reviewer 3 Report
The paper treats on an important aspect of remote monitoring of wheat diseases. However, I see many issues which should be amended before publication of this paper.
1. Materials and methods. This section should be renumbered from 4 to 2 and shifted in proper location of the paper - just after the 1. Introduction section. Consequently, tables and some references should be renumbered. Many improvements to Materials and Methods are needed. The authors should add short information about a climate (preferably according to Koppen-Geiger classification) and - optionally - soils (preferably soil texture and Soil Reference Group according to WRB 2015). The authors should add information about all seasons of studies, because in the caption for Table 2 „three growing seasons” are mentioned but in Table 3 only the measurement dates during the season 2022 (2022/23) are provided. There are also some other issues to be amended, such as lack of information of producers of equipment used in the study, as well as providers of the software used, the lack of definition of SBC (spectral brightness coefficient). How many records were used for statistical analyses? Did all test plots (10) were infected with all diseases, or each plot was infected with one disease? Perhaps graphical scheme of main plot and small-plot experiment would be helpful? Please, add information on dimension, area and the number of small plots. Actually, this section should be rewritten and divided in subsections, such as, for example: Location, climate and soil of experiment, Experimental design (including main and small-plot experiments, artificial infection, fungicide treatments etc), Data collection (assessment of plant damage, ground aerial and satellite measurements, air sampling of spores), Statistical analyses.
2. Conceptual issues. It is clear that visual indices, as NDVI may be used to monitor plant leaf disease, but how we can distinguish between low NDVI caused by insufficient N supply or other issue from low NDVI caused by plant disease? And, is this possible to distinguish particular plant leaf disease using remote sensing?
3. Designation of growth stages - the authors use the letter „Z” with respective number. However, it is more approppriate to design them with GS (Growth stage) or BBCH (Biologische Bundesanstalt, Bundessortenamt und CHemische Industrie, https://en.wikipedia.org/wiki/BBCH-scale) than Z letter. Although these scale come from Zadoks scale (letter Z), the acronims GS and BBCH are more commonly used in current scientific literature. Please, amend it in the whole text and all figures and tables!
4. Figures are quite difficult to interpret. I think they should be larger, especially the fonts. Perhapos the authors could hide particular dates in figures 2, 3, 4 and 5 (numbering error the figure 3 appears to times, when I mention figure 5 I refer to the figure just after line 177) and conserve the month and growth stages.
Consequently, I recommend major revision of this paper.
DETAILED
Lines 95, 282 and 352: I think the authors should replace plant list with plant leaf in some parts of the paper. Please, check it.
Figure 1. Does it refer to main test plots experiment or small-plot experiment?
Lines 103- 112: This paragraph fits more to Materials and methods section, than result. Anyway, I don’t agree with part about „complete and relatively continuous nature of data acquisition” of satellite imagery. Even, if we consider that the satellite flight over particular area every 5 days (Sentinel 2) is „relatively continuous” we must remember about the cloud cover. Additionally, we must remember that access to satellite data may be restricted. Current international situation is very insecure now.
Table 2. The lack of statistical significane of correlation between ground based spectrometry data and aerial survey.
Lines 247-250: I cannot agree that satellite imagery and surveys made by UAVs are „interchangeable”between them, and, as I can suppose, with ground spectrometric studies. It depend on the field, date and sensor used. Please, check the following papers:
Li et al 2023 Dynamic Changes and Influencing Factors of Vegetation in the “Green Heart” Zone of the Chang-Zhu-Tan Urban Agglomeration during the Past 21 Years
- https://www.mdpi.com/1660-4601/20/5/4517
Serrano et al. 2023 Pasture Quality Monitoring Based on Proximal and Remote Optical Sensors: A Case Study in the Montado Mediterranean Ecosystem
- https://www.mdpi.com/2624-7402/5/1/25
Farbo et al. 2022. Spectral Measures from Sentinel-2 Imagery vs Ground-Based Data from Rapidscan© Sensor: Performances on Winter Wheat - https://link.springer.com/chapter/10.1007/978-3-031-17439-1_15
Stettmer et al. 2022 Analysis of Nitrogen Uptake in Winter Wheat Using Sensor and Satellite Data for Site-Specific Fertilization - https://www.mdpi.com/2073-4395/12/6/1455
Mezera et al. 2021 Comparison of Proximal and Remote Sensing for the Diagnosis of Crop Status in Site-Specific Crop Management https://www.mdpi.com/1424-8220/22/1/19
Bausch and Khosla 2009 QuickBird satellite versus ground-based multi-spectral data for estimating nitrogen status of irrigated maize https://link.springer.com/article/10.1007/s11119-009-9133-1
and many others. After this the Introduction, Discussion and References should be quite extended.
Line 279: the phase BBCH/GS 30-32 should be renamed to „beginning of stem elongation”.
Lines 302: Please add information about the producer of ASD Field-Spec 3 Hi-Res spectroradiometer and define, if it works in active or passive mode.
Line 316: Please define SBC (spectral brightness coefficient) and add respective reference.
Line 319: Please add information about the producer of Parrott SEQUOIA+ multispectral camera and the altitude of the UAV’s flight over test plots.
Line 322: Please add information on provider of Pix4D software.
Line 330: Please add information about the producer of PSL-3 air sampler.
Line 338: Please, add particular link to the QGIS software.
Line 346: Please add information on provider of Statistica 2010 software.
Author Response
Responses to reviewer №3
We sincerely thank both reviewers for constructive and useful comments and have tried to address all of them. The corrections have been highlighted in yellow in the text of the paper for the ease of reference.
Our detailed responses are below.
Reviewer #3:
The paper treats on an important aspect of remote monitoring of wheat diseases. However, I see many issues which should be amended before publication of this paper.
Point 1: Materials and methods. This section should be renumbered from 4 to 2 and shifted in proper location of the paper - just after the 1. Introduction section. Consequently, tables and some references should be renumbered. Many improvements to Materials and Methods are needed. The authors should add short information about a climate (preferably according to Koppen-Geiger classification) and - optionally - soils (preferably soil texture and Soil Reference Group according to WRB 2015). The authors should add information about all seasons of studies, because in the caption for Table 2 „three growing seasons” are mentioned but in Table 3 only the measurement dates during the season 2022 (2022/23) are provided. There are also some other issues to be amended, such as lack of information of producers of equipment used in the study, as well as providers of the software used, the lack of definition of SBC (spectral brightness coefficient). How many records were used for statistical analyses? Did all test plots (10) were infected with all diseases, or each plot was infected with one disease? Perhaps graphical scheme of main plot and small-plot experiment would be helpful? Please, add information on dimension, area and the number of small plots. Actually, this section should be rewritten and divided in subsections, such as, for example: Location, climate and soil of experiment, Experimental design (including main and small-plot experiments, artificial infection, fungicide treatments etc), Data collection (assessment of plant damage, ground aerial and satellite measurements, air sampling of spores), Statistical analyses.
Response 1: All comments and suggestions were accepted, corrected and additions were made to the article.
Point 2. Conceptual issues. It is clear that visual indices, as NDVI may be used to monitor plant leaf disease, but how we can distinguish between low NDVI caused by insufficient N supply or other issue from low NDVI caused by plant disease? And, is this possible to distinguish particular plant leaf disease using remote sensing?
Response 2:
In the test plots, the main factor of heterogeneity was the level of disease development. The level of nitrogen nutrition should have been the same in the test plots, as fertilizers were applied uniformly. We did not put this question in our task, although we realize that it is correct and in the future we will take this into account as well. The work investigates the time period of active development of pathogens, which correlates well with this very factor (level of disease development). Of course, it cannot be excluded that small differences in nitrogen content of the plots markedly affect the coefficients. Therefore, it is planned to pay special attention to this aspect in future studies.
Without the use of hyperspectral data, it is probably impossible to distinguish a specific disease based only on the space imagery that we used. The effect of different diseases on the spectral characteristics of plants is similar and not differentiated when using the wide channels of modern multispectral imaging systems. However, the use of hyperspectral data has some potential for this. Unfortunately, the realization of this task is associated with the need for multiyear multitemporal studies on a large number of test plots and is complicated by field experimental conditions with poorly predictable side effects.
Point 3. Designation of growth stages - the authors use the letter „Z” with respective number. However, it is more approppriate to design them with GS (Growth stage) or BBCH (Biologische Bundesanstalt, Bundessortenamt und CHemische Industrie, https://en.wikipedia.org/wiki/BBCH-scale) than Z letter. Although these scale come from Zadoks scale (letter Z), the acronims GS and BBCH are more commonly used in current scientific literature. Please, amend it in the whole text and all figures and tables!
Response 3:
The text of the article has been updated in accordance with the comment.
Point 4. Figures are quite difficult to interpret. I think they should be larger, especially the fonts. Perhapos the authors could hide particular dates in figures 2, 3, 4 and 5 (numbering error the figure 3 appears to times, when I mention figure 5 I refer to the figure just after line 177) and conserve the month and growth stages.
Consequently, I recommend major revision of this paper.
DETAILED
Lines 95, 282 and 352: I think the authors should replace plant list with plant leaf in some parts of the paper. Please, check it.
Figure 1. Does it refer to main test plots experiment or small-plot experiment?
Lines 103- 112: This paragraph fits more to Materials and methods section, than result. Anyway, I don’t agree with part about „complete and relatively continuous nature of data acquisition” of satellite imagery. Even, if we consider that the satellite flight over particular area every 5 days (Sentinel 2) is „relatively continuous” we must remember about the cloud cover. Additionally, we must remember that access to satellite data may be restricted. Current international situation is very insecure now.
Table 2. The lack of statistical significane of correlation between ground based spectrometry data and aerial survey.
Lines 247-250: I cannot agree that satellite imagery and surveys made by UAVs are „interchangeable”between them, and, as I can suppose, with ground spectrometric studies. It depend on the field, date and sensor used.
Response 4:
The text of the article has been updated in accordance with the comment.
Point 5: Please, check the following papers:
Li et al 2023 Dynamic Changes and Influencing Factors of Vegetation in the “Green Heart” Zone of the Chang-Zhu-Tan Urban Agglomeration during the Past 21 Years
- https://www.mdpi.com/1660-4601/20/5/4517
Serrano et al. 2023 Pasture Quality Monitoring Based on Proximal and Remote Optical Sensors: A Case Study in the Montado Mediterranean Ecosystem
- https://www.mdpi.com/2624-7402/5/1/25
Farbo et al. 2022. Spectral Measures from Sentinel-2 Imagery vs Ground-Based Data from Rapidscan© Sensor: Performances on Winter Wheat - https://link.springer.com/chapter/10.1007/978-3-031-17439-1_15
Stettmer et al. 2022 Analysis of Nitrogen Uptake in Winter Wheat Using Sensor and Satellite Data for Site-Specific Fertilization - https://www.mdpi.com/2073-4395/12/6/1455
Mezera et al. 2021 Comparison of Proximal and Remote Sensing for the Diagnosis of Crop Status in Site-Specific Crop Management https://www.mdpi.com/1424-8220/22/1/19
Bausch and Khosla 2009 QuickBird satellite versus ground-based multi-spectral data for estimating nitrogen status of irrigated maize https://link.springer.com/article/10.1007/s11119-009-9133-1
and many others. After this the Introduction, Discussion and References should be quite extended.
Response 5:
Papers have been reviewed and added in the introduction and discussion of the article
Point 6:
Line 279: the phase BBCH/GS 30-32 should be renamed to „beginning of stem elongation”.
Lines 302: Please add information about the producer of ASD Field-Spec 3 Hi-Res spectroradiometer and define, if it works in active or passive mode.
Line 316: Please define SBC (spectral brightness coefficient) and add respective reference.
Line 319: Please add information about the producer of Parrott SEQUOIA+ multispectral camera and the altitude of the UAV’s flight over test plots.
Line 322: Please add information on provider of Pix4D software.
Line 330: Please add information about the producer of PSL-3 air sampler.
Line 338: Please, add particular link to the QGIS software.
Line 346: Please add information on provider of Statistica 2010 software.
Response 6:
The text of the article has been updated in accordance with the comment

Round 2
Reviewer 1 Report
The response by the authors is really poor.
Normally authors should response point to point and not say to the referee "check the yellow parts"
So I went through the ywllo (and not yellow) parts and found no answer to my comments. In particular:
Major comments
Authors write that "Ten test plots were allocated to create an artificial infectious background and, accordingly, 10 control plots to ensure comparability of aerospace survey data with the results of ground-based spectrometric measurements within the experimental field with a total area of 1 ha. The size of each test plot was 10 × 10 m (100 m2)."
This is unclear: 10 test plots + 10 control plots = 20 plots x 100 m2= 2000 m2. What about the remaining area (8000 m2)? How was it used? Not used at all? Why not using it as further control? or as middle infection degree area?
Please add ground resolution/total area for ground measurment
Satellite data have a ground resolution of 0.5 m: how were plots borders considered? were they excluded from analyses? Please provide more information on data processing.
Has the small plot ben analysd also by the sallite?
Please provide more information on how the actual infection of plots have been evaluated.
Minor comments
Remove quotation marks from the title
Improve readability of fonts in the graphs/figures (increase font size, without increasing the total dimension of graphs)
"Space" or "aerospace imaging" terms are not commonly used in agriculture: "remote sensing" or "satellite" is much more common
Avoid using "we": preferred using impersonal form
Pleasee provide an answer to each point above.
English needs a minor review
Author Response
Reviewer #1:
Major comments
Comment 1.
Authors write that "Ten test plots were allocated to create an artificial infectious background and, accordingly, 10 control plots to ensure comparability of aerospace survey data with the results of ground-based spectrometric measurements within the experimental field with a total area of 1 ha. The size of each test plot was 10 × 10 m (100 m2)."
This is unclear: 10 test plots + 10 control plots = 20 plots x 100 m2= 2000 m2. What about the remaining area (8000 m2)? How was it used? Not used at all? Why not using it as further control? or as middle infection degree area?
Answer 1.
The remaining area is the background areas, which avoid errors associated with field edge effects. Figure 5 A, B shows the control and infected plots of the main experiment. Since the influence of the edge effect on the control plots is less significant, they were placed without buffer sections between them.
(see lines 478-484)
Comment 2.
Please add ground resolution/total area for ground measurment
Answer 2.
Ground-based spectrometry was carried out non-contact at a height of 1.2 - 1.4 m from the earth's surface in the electromagnetic radiation range from 350 to 2500 nm with a spectral resolution of 1-10 nm. To do this we used the ASD FieldSpec 3 Hi-Res spectro-radiometer [55] designed to measure the absolute and relative values of the radiance. We carried out measurements in clear sunny weather with a minimum amount of clouds at sun heights of more than 35˚. This was done to ensure the comparability of the obtained data. Under such conditions lighting conditions change much less which reduces the error associated with the influence of this factor. Vegetation measurements in the small-plot experiment were carried out in two series of five repetitions which were in-terrupted by measurements of the calibration white panel. This measure was taken to reduce the influence of the uneven lighting factor. For each plot of the main experiment 30 measurements of the vegetation cover were carried out as well as measurements of the white calibration panel at the beginning and end of each series. Vegetation cover was measured from one corner of the site to the opposite corner in accordance with the pro-cedure for conducting field inspections of plants for the presence of pathogens. The area of one measurement covered by the spectroradiometer sensor was 0.222 m2. This can be considered as the spatial resolution of ground measurements.Thus, the area of measurements of each test plot of the main experiment was about 7 m2. and for the small-plot experiments - 2.22 m2. The total measurement area of all test plots was about 60 m2.
(see lines 560-583).
Comment 3.
Satellite data have a ground resolution of 0.5 m: how were plots borders considered? were they excluded from analyses? Please provide more information on data processing.
Answer 3.
When extracting spectral brightness values. edge pixels were excluded to avoid mixed pixels. The test sites were marked on the ground with the help of special markers. This allowed the angles of the sites to be determined. Spectra inside these sites were taken with 1 m indentation inside the boundaries.
(see lines 615-618).
Comment 4.
Has the small plot ben analysd also by the sallite?
Answer 4.
Unfortunately, both the area of the small-plot experiment and edge effect influence, prevented obtaining valid data. Therefore, the satellite data analysis for the small experiment was not carried out.
(see lines 604-607).
Comment 5.
Please provide more information on how the actual infection of plots have been evaluated.
Answer 5.
The assessment of the degree of disease development was based on the visual counting of the ratio of the proportion of the affected area of the plant leaf lamina to its total area. Visual counts of winter wheat disease development were carried out along the diagonal of each plot with an area of 10 m2. During the surveys 30 plants were selected then for each tier (first. second leaf. etc.) according to international scales. the percentage of leaf lesions was given. The Peterson scale [51] was used to assess the degree of rust disease damage. the modified Saari and Prescot scale [52] was used to assess the degree of pyrenophorosis damage. and special scales developed by CIMMYT [53] were used to assess the degree of powdery mildew. spot blight and septoriosis damage.
For each test plot. the average indices of the degree of disease development were calculated according to formula (1) (Table 9-10):
(1)
where R is the average degree of development of the disease. %;
r is the degree of development of the disease of an individual plant. %;
n is the total number of registered plants. pcs.;
(see lines 526-543).
Minor comments
Comment 1.
Remove quotation marks from the title
Answer 1.
The quotation marks have been removed
(see lines 2-3).
Comment 2.
Improve readability of fonts in the graphs/figures (increase font size, without increasing the total dimension of graphs)
Answer 2.
The readability has been removed
(see lines 140-141; 158-160; 169-170; 184-185).
Comment 3.
"Space" or "aerospace imaging" terms are not commonly used in agriculture: "remote sensing" or "satellite" is much more common
Answer 3.
Terminology has been adjusted throughout the text
(see lines 135, 138, 152, 162, 256, 266, 349, 441, 450, 471, 611, 659)
Comment 4.
Avoid using "we": preferred using impersonal form
Answer 4.
Corrected
Response:
All comments and suggestions were accepted, corrections and additions were made to the article.

Reviewer 2 Report
The authors did some work, but failed to respond to every comment in the letter. If they would please provide explicit pointed explanation of each revision. Although they say 'all comments accepted' there are a number which were not responded to. For example, I made some suggestions for the discussion, but no changes seem to have been made in this section.
With new explanations in the methods come new questions. For example, Figure 5 now does not appear to be connected to new Tables 8, 9 and 10. Could the plots be labeled? It is not clear which experiment these new tables refer to. Tables 6 and 7 need to be further explained - which experiment do these refer to? I think now there is too much detail instead of too little.
Finally, why the methods after the results? Is this typical with this journal?
It is in English - language is very complex though, making the material less accessible.
Author Response
Reviewer #2:
Comment 1
The authors did some work, but failed to respond to every comment in the letter. If they would please provide explicit pointed explanation of each revision. Although they say 'all comments accepted' there are a number which were not responded to. For example, I made some suggestions for the discussion, but no changes seem to have been made in this section.
Answer 1
We apologize for any sloppiness with regard to the responses to comments and consider it appropriate to detail the responses to comments from the first review:
Lines 54-56 - what does it mean ‘environmental factors’ on ‘data acquisition’ – do you mean acquisition of satellite data? If so, what environmental factors do you mean? Also for ‘transformation of models’ transferred to a ‘new index system’ – do you mean when you are applying the model to a new region? If so please say that. Otherwise, please define ‘new index system’ – what new index?
Answer Another problem is the difficulty of taking into account the mutual influence of environmental factors (soil type, weather conditions, illumination of the earth's surface, etc,), which determine the conditions for obtaining experimental data within a particular research region, The influence of these factors on the spectral characteristics of agricultural crops can be non-uniform during the growing season, This requires a significant transformation of models created in specific conditions when applied to a new region [13,23-25], Lines 55-61.
Lines 61-62 – by cultivar, I assume you mean different varieties of wheat – please state that. Cultivar means several different things and can be ambiguous in a diverse disciplinary community.
Answer The difference in the spectral responses of different varieties of the same crop also presents significant difficulties Therefore, the model for diagnosing the development of the disease in one particular wheat variety may not be applicable to another variety [32-35], Lines 65-68.
Line 71 – time series of data of what? Disease prevalence? Or remote sensing time series?
Answer The last but not least is the need to study the dynamics of changes in the spectral images of cultivated crops against the background of the development of diseases over time during their growing season [10], Lines 68-71.
Please put the Materials and Methods section before the Results section – not sure why it is out of order here.
Answer Sections of the manuscript have been placed according to the requirements of the journal Plants. https://www.mdpi.com/journal/plants/instructions
Table 3 – this is confusing – I think it would be better to put Field inspections in the column on the left and the other data acquisitions on the right.
Answer Table 3 has been restructured to reflect the comment. It is now table 8. Lines 505-506.
Table 3 – by ‘space photography’ do you mean the ‘ground-based spectrometry’ mentioned in the next paragraph, the ‘aerial photography’ discussed on line 319, or the Planet labs observations mentioned in line 323? Please be explicit here – tell us EXACTLY what data you are talking about.
Answer The content of the table has been modified in accordance with the comment. Lines 505-506.
Line 98, Figure 1 – how is the ‘average progression of the disease, %’ calculated? It does not seem to be presented in the methods section. Please provide that – is it from the standard protocols you mentioned 283-284? If so, please state exactly how you calculated the percentage, as the ‘standard protocols’ are quite unknown to the remote sensing community, who should be interested in citing this paper if it is clear.
Answer The assessment of the degree of disease development was based on the visual counting of the ratio of the proportion of the affected area of the plant leaf lamina to its total area. Visual counts of winter wheat disease development were carried out along the diagonal of each plot with an area of 10 m2. During the surveys. 30 plants were selected. then for each tier (first. second leaf. etc.). according to international scales. the percentage of leaf lesions was given. The Peterson scale [51] was used to assess the degree of rust disease damage. the modified Saari and Prescot scale [52] was used to assess the degree of pyrenophorosis damage. and special scales developed by CIMMYT [53] were used to assess the degree of powdery mildew. spot blight and septoriosis damage.
For each test plot. the average indices of the degree of disease development were calculated according to formula (1) (Table 9-10):
(1)
where R is the average degree of development of the disease. %;
r is the degree of development of the disease of an individual plant. %;
n is the total number of registered plants. pcs.;
(see lines 526-541).
Lines 113-120 – I think these are methods and should be put in a methods section, explaining how the correlations were done. Also, please explain the ‘aerial survey’ in the methods– is this the sampling of spores in the air above the crop that was mentioned in the introduction?
Answer
In parallel with the accounting of the degree of disease development. air sampling over winter wheat crops was carried out using the original air sampler PSL-3 developed at FSBSI FRCBPP [54]. The device is an impactor. inside which there is a slide with the initial size of the composition (vaseline) in which the spores of phytopathogenic fungi are de-posited. Sampling was conducted along the diagonal of each plot at five points. The sam-pling time was one minute. To detect. identify and quantify phytopathogenic fungi. the samples were examined under a light microscope at 10x objective magnification. Lines 551-557.
Correlation analysis of ground-based spectrometry and aerial survey data for May 27-28 (the period where both types of surveys were carried out) showed the following: despite the different density of repeated measurements within the main plots and areas of the small-plot experiment (0.3 meas./m2 versus 5 meas./ m2). a stable correlation of the obtained values can be traced only for the red and IR regions of the spectrum (Table 11). This suggests that differences in the pathogenic background can be reliably detected using UAVs only in these ranges. This also suggests that it is not necessary to strive for a high density of measurements per unit area in order to conduct ground-based spectrometry.
Correlation analysis of the relationship between spectral data and disease records was carried out using the SciPy library of the Python programming language [https://scipy.org/]. Correlation analysis of the relationship between the disease development and air pollution indicator was carried out on the basis of non-parametric statistics methods using the Spearman test at a high 95% significance level using the Statistiсa 2010 program. Lines 619-637.
Line 122 – what is SBC? Acronym does not appear to be defined.
Answer See Data obtained from ground-based spectrometric measurements are a set of spectral brightness coefficient (SBC) values that indicate the degree to which sunlight is reflected from plant surfaces at each wavelength. This data was processed automatically using a spe-cially written script in the Python programming language. Lines 579-582.
Table 1 – what are the two things being correlated here? Why don’t you present the ground-based infection rates instead of spectrometry data? It is hardly an interesting conclusion that the satellite data is able to capture the same information as the spectrometer on the ground – this has been shown hundreds of times in the past forty years in the literature. I would just delete this unless you can present information regarding the fungal infection instead of merely reflectance.
Answer Table 1 has been moved to the Materials and Methods section and renumbered as Table 11 in accordance with previous recommendations. Lines 630-631.
Figure 2 – what are the numbers on the X axis? Dates? Please provide a label of the axis on this plot. Also, again the SBC label instead of ‘index value’ for the different spectral indices. I assume that for the first three figures is spectral reflectance, calculated from the Planet imagery, but it is not clear. Please revise the caption of this figure to specify ‘Planet imagery’ and define SBC.
Answer Figure 2 has been finalized in accordance with the comment.
Lines 247-250 – interesting conclusion that very high resolution (VHR) imagery would be ‘less expensive’ than ground spectrometry. There are a lot of additional costs for acquiring, processing, and delivering VHR imagery beyond simply purchasing it, which include having access to technical expertise. Also please explain what you mean by ‘less expensive analogues of spectrometric studies.’ – This I don’t get. What studies?
Answer The low productivity of ground-based spectrometry relative to satellite and UAV data was noted. At the same time, the multitemporal dynamics of the spectral image of plants for each of these survey levels remains similar. In this regard, in such studies it is recommended to focus on aerial and satellite based data, especially if there is a task of scaling. However, since the ground spectrometry is based on active sensors with their own light source and is therefore less dependent on weather, it should not be abandoned entirely. At the current stage, ground-based spectrometry is used as a method that allows one to study in detail the changes both in the spectral characteristics of crops and over time in order to identify certain patterns. These data can be used to better interpret the results of space and unmanned surveys. Lines 421-431.
Line 263 - Additional discussion points that could be added here include how predictive the authors think this system is. Can the satellite data identify an emerging problem sufficiently early so that the farmer could intervene in the development of the fungal pathogen before yield impacts occur? This is the central question that the paper should address.
Answer We have tried to answer these questions by supplementing the text of the discussion section. Lines 353-420.
Comment 2
With new explanations in the methods come new questions. For example, Figure 5 now does not appear to be connected to new Tables 8, 9 and 10. Could the plots be labeled? It is not clear which experiment these new tables refer to. Tables 6 and 7 need to be further explained - which experiment do these refer to? I think now there is too much detail instead of too little.
Answer 2
The Tables indicated which experiment they illustrate. The scheme of the experiment was revised taking into account the recommendations.
(see lines 315, 324, 328; 476-488; 507, 546, 550)
Comment 3
Why the methods after the results? Is this typical with this journal?
Answer 3
This is a requirement put forward by the journal for publication. https://www.mdpi.com/journal/plants/instructions
Response: All comments and suggestions were accepted, corrections and additions were made to the article.

Reviewer 3 Report
GENERAL
The paper improved considerably.
Particularly, more extensive description of methods increased the quality of the paper.
I recommend publication of these paper after rather small and editorial corrections.
DETAILED
Please, be consistent in designation of growth stages (phenophases) and consequently use codes beginning form „GS” rather than „BBCH” or „Z” throughout the text, tables and figures! For example in lines 91-92 „BBCH” is used, and in the tables 1-7, and lines 226, 235 „GS” is present, while in the figures 1-4 „Z” was still conserved. As I can note, the „GS” is most frequently used.
Please, increase phonts in all figures - they are difficult to read at the 100% zoom of the document.
Lines 79 and 83: Please, check, the reference [36] is described as Serrano et al. (2023) in the text and as Bausch and Khosla (2010) in reference list, and vice versa.
Line 321: Please, add missing reference number at the end of the phrase.
Lines 334-347: I have certain doubts regarding recommendation to „... abandon ground -based spectrometry ...”, although it could be quite limited. Remember, that ground spectrometry is based on active sensores, with their own light source and thus less dependent on weather (mainly clouds) which limit the quality of the UAV-, air- and satellite-based data. Please remember, that some weather phenomena - cloud cover, rain, wind etc may make impossible the obtention of UAV, air and satellite data.
Lines 355-370: I recommend to shorten slightly the part of the text and prepare a diagram showing the monthly average temperatures and sum of precipitations during the study. It may help in data interpretation and future discussion in other studies.
Line 410: Please replace „the second growing season” with the „the second time period”, because the „season” suggests other year.
Line 472: Please correct small errot in „spectroradirometer sensor”.
Author Response
Reviewer #3:
Comment 1
Please, be consistent in designation of growth stages (phenophases) and consequently use codes beginning form „GS” rather than „BBCH” or „Z” throughout the text, tables and figures! For example in lines 91-92 „BBCH” is used, and in the tables 1-7, and lines 226, 235 „GS” is present, while in the figures 1-4 „Z” was still conserved. As I can note, the „GS” is most frequently used.
Response 1
Corrected
(see lines 92, 240, 250, 259, 265, 277, 287, 291, 354, 375, 380, 389, 398, 410, 510, 514, 520, Table 1-10; figures 1-4)
Comment 2
Please, increase fonts in all figures - they are difficult to read
Response 2
Corrected
Comment 3
Lines 79 and 83: Please, check, the reference [36] is described as Serrano et al. (2023) in the text and as Bausch and Khosla (2010) in reference list, and vice versa.
Response 3
Corrected
(see lines 79, 83)
Comment 4
Line 321: Please, add missing reference number at the end of the phrase.
Response 4
The link has been inserted, it is now line 344.
Comment 5
Lines 334-347: I have certain doubts regarding recommendation to „... abandon ground -based spectrometry ...”, although it could be quite limited. Remember, that ground spectrometry is based on active sensores, with their own light source and thus less dependent on weather (mainly clouds) which limit the quality of the UAV-, air- and satellite-based data. Please remember, that some weather phenomena - cloud cover, rain, wind etc may make impossible the obtention of UAV, air and satellite data.
Response 5
Thanks for the comment. The text has been replaced with the following:
The low productivity of ground-based spectrometry relative to satellite and UAV data was noted. At the same time, the multitemporal dynamics of the spectral image of plants for each of these survey levels remains similar. In this regard, in such studies it is recommended to focus on aerial and satellite based data, especially if there is a task of scaling. However, since the ground spectrometry is based on active sensors with their own light source and is therefore less dependent on weather, it should not be abandoned entirely. At the current stage, ground-based spectrometry is used as a method that allows one to study in detail the changes both in the spectral characteristics of crops and over time in order to identify certain patterns. These data can be used to better interpret the results of space and unmanned surveys.
(see lines 421-430)
Comment 6
Lines 355-370: I recommend to shorten slightly the part of the text and prepare a diagram showing the monthly average temperatures and sum of precipitations during the study. It may help in data interpretation and future discussion in other studies.
Response 6
Thanks for the comment. We agree that such a diagram would be beneficial to the article representaion. Unfortunately, we do not have the opportunity to compile it in a short time. But we will definitely consider it in future studies.
Comment 7
Line 410: Please replace „the second growing season” with the „the second time period”, because the „season” suggests other year.
Response 7
Corrected
(see lines 513)
Comment 7
Line 472: Please correct small errot in „spectroradirometer sensor”.
Response 7
Corrected
(see lines 575)с

Round 3
Reviewer 1 Report
My comments this time have been properely addressed and discussed, and the paper accordingly improved.
For this reason I blieve the paper is now reeady for publication.
English is fine